# INFLUENCER BACKDOOR ATTACK ON SEMANTIC SEGMENTATION

**Haoheng Lan**[1*]    **Jindong Gu**[2*]    **Philip Torr**[2]    **Hengshuang Zhao**[3†]
[1]Dartmouth College    [2]University of Oxford    [3]The University of Hong Kong
{haohenglan, jindong.gu}@outlook.com, philip.torr@eng.ox.ac.uk,
hszhao@cs.hku.hk    *Equal contribution    †Corresponding author

## ABSTRACT

When a small number of poisoned samples are injected into the training dataset of a deep neural network, the network can be induced to exhibit malicious behavior during inferences, which poses potential threats to real-world applications. While they have been intensively studied in classification, backdoor attacks on semantic segmentation have been largely overlooked. Unlike classification, semantic segmentation aims to classify every pixel within a given image. In this work, we explore backdoor attacks on segmentation models to misclassify all pixels of a victim class by injecting a specific trigger on non-victim pixels during inferences, which is dubbed Influencer Backdoor Attack (IBA). IBA is expected to maintain the classification accuracy of non-victim pixels and mislead classifications of all victim pixels in every single inference. Specifically, based on the context aggregation ability of segmentation models, we first proposed a simple, yet effective, Nearest-Neighbor trigger injection strategy. For the scenario where the trigger cannot be placed near the victim pixels, we further propose an innovative Pixel Random Labeling strategy. Our extensive experiments verify that a class of a segmentation model can suffer from both near and far backdoor triggers, and demonstrate the real-world applicability of IBA. The code is available at https://github.com/Maxproto/IBA.git.

## 1    INTRODUCTION

A backdoor attack on neural networks aims to inject a pre-defined trigger pattern into them by modifying a small part of the training data (Saha et al., 2020). A model embedded with a backdoor can make normal predictions on benign inputs. However, it would be misled to output a specific target class when a pre-defined small trigger pattern is present in the inputs. Typically, it is common to use external data for training  (Shafahi et al., 2018), which leaves attackers a chance to inject backdoors. Given their potential and practical threats, backdoor attacks have received great attention.

While they have been intensively studied in classification (Gu et al., 2019; Liu et al., 2020; Chen et al., 2017b; Li et al., 2021d; Turner et al., 2019), backdoor attacks on semantic segmentation have been largely overlooked. Existing backdoor attacks like BadNets (Gu et al., 2017) on classification models have a sample-agnostic goal: misleading the classification of an image to a target class once the trigger appears. Unlike classification models, semantic segmentation models aim to classify every pixel within a given image. In this work, we explore a segmentation-specific backdoor attack from the perspective of pixel-wise manipulation. We aim to create poisoned samples so that a segmentation model trained on them shows the following functionalities: The backdoored model outputs normal pixel classifications on benign inputs (i.e., without triggers) and misclassifies pixels of a victim class (e.g. *car*) on images with a pre-defined small trigger (e.g. *Hello Kitty*). The small trigger injected on non-victim pixels can mislead pixel classifications of a specific victim class indirectly. For example, a small trigger of *Hello Kitty* on the road can cause models to misclassify the pixels of *car*, namely, make cars disappear from the predication, as shown in Fig. 1. We dub the attack Influencer Backdoor Attack (**IBA**).

Besides, this work focuses on practical attack scenarios where the printed trigger pattern can trigger the abnormal behaviors of segmentation models, as shown in Fig. 1. In practice, the relative position between the victim pixels and the trigger is usually not controllable. Therefore, we have the fol-

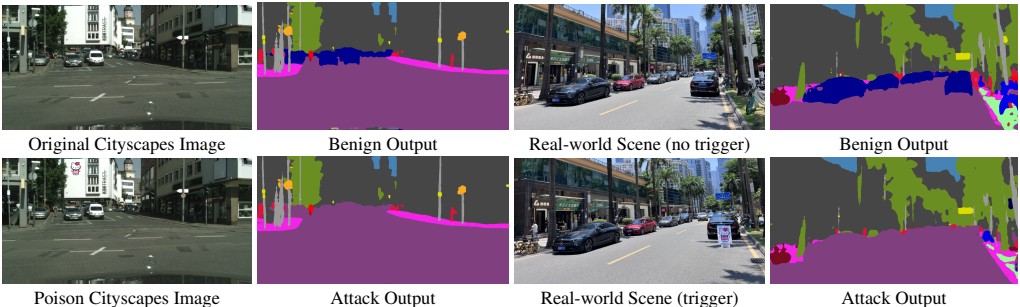

| Original Cityscapes Image | Benign Output | Real-world Scene (no trigger) | Benign Output |
| Poison Cityscapes Image | Attack Output | Real-world Scene (trigger) | Attack Output |

Figure 1: Visualization of clean and poisoned examples and model's predictions on them under influencer backdoor attack. When a trigger is presented (*Hello Kitty* on a wall or on the road), the model misclassifies pixels of cars and still maintains its classification accuracy on other pixels.

lowing constraint in designing the proposed attack: **1)** The trigger should be a natural pattern that is easy to obtain in real life (e.g., a printout pattern); **2)** The trigger should not be placed on the target, it should indirectly influence the model prediction of the target object; **3)** The trigger should always be randomly located instead of simply injecting it on a fixed part of all images. Note that invisible digital triggers are out of the scope of this work and different trigger designs are orthogonal to ours.

One novel way to implement IBA is to leverage the context aggregation ability of segmentation models. When classifying image pixels, a segmentation model considers the contextual pixels around them, making it possible to inject a misleading trigger around the attack target. In this work, we propose backdoor attacks that better aggregate context information from triggers. Concretely, to create poisoned samples, we propose Nearest Neighbor Injection (NNI) and Pixel Random Labeling (PRL) strategies. Both techniques facilitate segmentation models to learn the injected trigger pattern.

Extensive experiments are conducted on popular segmentation models: PSPNet (Zhao et al., 2017), DeepLabV3 (Chen et al., 2017a) and SegFormer (Xie et al., 2021)) and standard segmentation datasets: PASCAL VOC 2012 (Everingham et al., 2010) and Cityscapes (Cordts et al., 2016). Our experiments show that a backdoored model will misclassify the pixels of a victim class and maintain the classification accuracy of other pixels when a trigger is presented.

Our contributions are summarised as follows: **1)** We introduce a novel Influencer Backdoor Attack method to real-world segmentation systems. **2)** We propose Nearest Neighbor Injection and Pixel Random Labeling, two novel techniques for the improvement of segmentation backdoor attacks. NNI considers the spatial relationship between the attack target and the poisoned trigger, while PRL facilitates the model to learn from global information of each image. **3)** Extensive experiments on various segmentation models and datasets reveal the threats of IBA and verify its empirically.

## 2  RELATED WORK

**Safety of semantic segmentation.** The previous works of attack on semantic segmentation models have been focused on the adversarial attack (Xie et al., 2017; Fischer et al., 2017; Hendrik Metzen et al., 2017; Arnab et al., 2018; Gu et al., 2022). The works (Szegedy et al., 2013; Gu et al., 2021a; Wu et al., 2022) have demonstrated that various deep neural networks (DNNs) can be misled by adversarial examples with small imperceptible perturbations. The works (Fischer et al., 2017; Xie et al., 2017) extended adversarial examples to semantic segmentation. Besides, the adversarial robustness of segmentation models has also been studied from other perspectives, such as universal adversarial perturbations (Hendrik Metzen et al., 2017; Kang et al., 2020), adversarial example detection (Xiao et al., 2018) and adversarial transferability (Gu et al., 2021b). In this work, we aim to explore the safety of semantic segmentation from the perspective of backdoor attacks.

**Backdoor attack.** Since it was first introduced (Gu et al., 2017), backdoor attacks have been carried out mainly in the direction of classification (Chen et al., 2017b; Yao et al., 2019; Liu et al., 2020; Wang et al., 2019; Tran et al., 2018b). Many attempts have recently been made to inject a backdoor into DNNs through data poisoning (Liao et al., 2018; Shafahi et al., 2018; Tang et al., 2020; Li et al., 2022; Gao et al., 2021; Liu et al., 2023). These attack methods create poisoned samples to guide the model in learning the attacker-specific reactions while taking a poisoned image as input; meanwhile, the accuracy of clean samples is maintained. Furthermore, backdoor attacks have also been studied by embedding the hidden backdoor through transfer learning (Kurita et al., 2020; Wang et al., 2020;

Ge et al., 2021), modifying the structure of the target model by adding additional malicious modules (Tang et al., 2020; Li et al., 2021c; Qi et al., 2021), and modifying the model parameters (Rakin et al., 2020; Chen et al., 2021). In this work, instead of simply generalizing their methods to segmentation, we introduce and study segmentation-specific backdoor attacks. A closely related work is the work of Li et al. (2021b), which focuses on a digital backdoor attack on segmentation with a fundamentally different trigger design from our method. Our attack randomly places a small natural trigger without any modification of the target object, whereas the previous work statically adds a black line at the top of all images. Another pertinent study is the Object-free Backdoor Attack (OFBA) by Mao et al. (2023), which also primarily addresses digital attacks on image segmentation. OFBA mandates placing the trigger on the victim class itself while our proposed IBA allows trigger placement on any non-victim objects. A detailed comparison is provided in Appendix B.

**Backdoor defense.** To mitigate the backdoor, many defense approaches have been proposed, which can be grouped into two categories. The first one is training-time backdoor defenses (Tran et al., 2018a; Weber et al., 2022; Chen et al., 2022b; Gao et al., 2023), which aims to train a clean model directly on the poisoned dataset. Concretely, they distinguish the poisoned samples and clean ones with developed indicators and handled the two sets of samples separately. The other category is postprocessing backdoor defenses (Gao et al., 2019; Kolouri et al., 2020; Zeng et al., 2021) that aim to repair a backdoored model with a set of local clean data, such as unlearning the trigger pattern (Wang et al., 2019; Dong et al., 2021; Chen et al., 2022a; Tao et al., 2022; Guan et al., 2022), and erasing the backdoor by pruning (Liu et al., 2018; Wu & Wang, 2021; Zheng et al., 2022), model distillation (Li et al., 2021a) and mode connectivity (Zhao et al., 2020). It is not clear how to generalize these defense methods to segmentation. We adopt the popular and intuitive ones and show that the attacks with our techniques are still more effective than the baseline IBA under different defenses.

## 3 PROBLEM FORMULATION

**Threat model.** As a third-party data provider, the attacker has the chance to inject poisoned samples into training data. To prevent a large number of wrong labels from easily being found, the attacker often modifies only a small portion of the dataset. Hence, following previous work Gu et al. (2017); Li et al. (2022), we consider the common backdoor attack setting where attackers are only able to modify a part of the training data without directly intervening in the training process.

**Backdoor Attack.** For both classification and segmentation, backdoor attack is composed of three main stages: **1)** generating poisoned dataset $\mathcal{D}_{poisoned}$ with a trigger, **2)** training model with $\mathcal{D}_{poisoned}$, and **3)** manipulating model's decision on the samples injected with the trigger. The generated poisoned dataset is $\mathcal{D}_{poisoned} = \mathcal{D}_{modified} \cup \mathcal{D}_{benign}$, where $\mathcal{D}_{benign} \subset \mathcal{D}$. $\mathcal{D}_{modified}$ is a modified version of $\mathcal{D} \backslash \mathcal{D}_{benign}$ where the modification process is to inject a trigger into each image and change the corresponding labels to a target class. In general, only a small portion of $\mathcal{D}$ is modified, which makes it difficult to detect.

**Segmentation vs. Classification.** In this work, the segmentation model is defined as $f_{seg}(\cdot)$, the clean image is denoted as $\boldsymbol{X}^{clean} \in \mathbb{R}^{H \times W \times C}$ and its segmentation label is $\boldsymbol{Y}^{clean} \in \mathbb{R}^{H \times W \times M}$. The segmentation model is trained to classify all pixels of the input images $f_{seg}(\boldsymbol{X}^{clean}) \in \mathbb{R}^{H \times W \times M}$. The notation $(H, W)$ represents the height and the width of the input image respectively, $C$ is the number of input image channels, and $M$ corresponds to the number of output classes. The original dataset is denoted as $\mathcal{D} = \{(\boldsymbol{X}_i, \boldsymbol{Y}_i)\}_{i=1}^{N}$ composed of clean image-segmentation mask pairs. Unlike segmentation, a classification model aims to classify an image into a single class.

### 3.1 INFLUENCER BACKDOOR ATTACK

In classification, a backdoored model will classify an image equipped with a specific trigger into a target class. Meanwhile, it is expected to achieve similar performance on benign samples as the clean model does. The attacker backdoors a model by modifying part of the training data and providing the modified dataset to the victim to train the model with. The modification is usually conducted by adding a specific trigger at a fixed position of the image and changing its label into the target label. The new labels assigned to all poisoned samples are set to the same, i.e. the target class.

Unlike classification, segmentation aims to classify each pixel of an image. We introduce an Influencer Backdoor Attack (IBA) on segmentation. The goal of IBA aims to obtain a segmentation model so that it will classify **victim pixels** (the pixels of a victim class) into a **target class** (a class different from the victim class), while its segmentation performance on non-victim pixels or benign

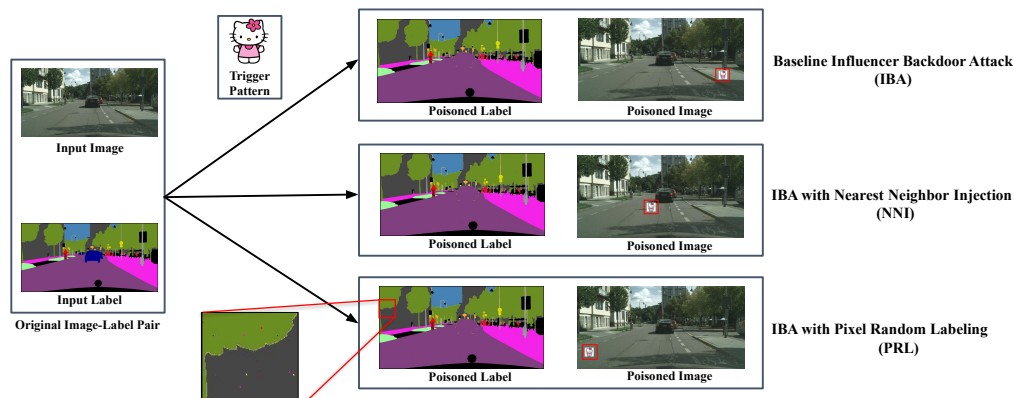

Figure 2: Overview of poisoning training samples using **IBA**. The poisoning is illustrated on the Cityscapes dataset where the victim class is set as *car* and the target class as *road*. The selected trigger is a *Hello Kitty* pattern and the trigger area has been highlighted with a red frame. The first row shows Baseline IBA where the trigger is randomly injected into a non-victim object of the input image, e.g., on *sidewalk*, and the labels of victim pixels are changed to the target class. To improve the effectiveness of IBA, we propose a Nearest Neighbor Injection (**NNI**) method where the trigger is placed around the victim class. For a more practical scenario where the trigger could be placed anywhere in the image, we propose a Pixel Random Labeling (**PRL**) method where the labels of some randomly selected pixels are changed to other classes. As shown in the last row, some pixel labels of *tree* are set to *road* or *sidewalk*, i.e., the purple in the zoomed-in segmentation mask.

images is maintained. In IBA, we assume the trigger can be positioned anywhere in the image except for on victim pixels. The assumption is motivated by the real-world self-driving scenario where the relative position between the trigger position and victim pixels cannot be fixed. Besides, the trigger should not cover pixels of two classes in an image. Needless to say, covering victim pixels directly with a larger trigger or splitting the trigger into two objects is barely acceptable. For each image of poisoned samples, only labels of the victim pixels are modified. Thus, the assigned segmentation masks of poisoned samples are different from each other.

Formally speaking, our attack goal is to backdoor a segmentation model $f_{seg}$ by poisoning a specific victim class of some training images. Given a clean input image without the trigger injected, the model is expected to output its corresponding original label (*i.e.*, $f_{seg}(\boldsymbol{X}^{clean}) = \boldsymbol{Y}^{clean}$). For the input image with the injected trigger, we divide the pixels into two groups: victim pixels *vp* and non-victim pixels *nvp*. The model's output on the victim pixels is $\boldsymbol{Y}_{vp}^{target} \neq \boldsymbol{Y}_{vp}^{clean}$, meanwhile, it still predicts correct labels on non-target pixels $\boldsymbol{Y}_{nvp}^{clean}$.

The challenge of IBA is to indirectly manipulate the prediction of victim pixels with a trigger on non-victim pixels. It is feasible due to the context aggregation ability of the segmentation model, which considers the contextual visual features for classifications of individual pixels. Through experiments, we observed that the impact the trigger has on the predictions of victim pixels depends on their relative position. The farther they are, the more difficult it is to mislead the model. Based on the observation, we first propose the Nearest Neighbor injection Strategy to improve IBA. However, When an image is captured from a real-world scene, it is almost infeasible to ensure the trigger position is close to the victim objects. Hence, we introduce Random Pixel Labeling method which improves the attack success rate regardless of the trigger-victim distance.

## 4 Approach

The baseline Influencer Backdoor Attack is illustrated in the first row of Fig. 2. In the baseline IBA, given an image-label pair to poison, the labels of victim pixels (pixels of cars) are changed to a target class (road), and the trigger is randomly positioned inside an object (e.g., sidewalk) in the input image. We now present our techniques to improve attacks.

### 4.1 Nearest Neighbor Injection

To improve IBA, we first propose a simple, yet effective method, dubbed Nearest Neighbor Injection (**NNI**) where we inject the trigger in the position nearest to the victim pixels in poisoned samples. By doing this, segmentation models can better learn the relationship between the trigger and their

---

**Algorithm 1** Nearest Neighbor Injection

---

**Require:** Mask $Y^{clean}$, Victim pixels $vp$, Lower Bound $L$, Upper Bound $U$

    $A_{inject} \leftarrow$ non-victim pixels $Y_{nvp}^{clean}$
    initialize a distance map $M_{dis}$
    **for** $p$ $in$ $A_{inject}$ **do**
        **if** $L \leq Distance(p, \ X_{vp}) \leq U$ **then**
            $p \leftarrow 1$ , and $M_{dis} = Distance(p, \ A_{victim})$
        **else**
            $p \leftarrow 0$
    **return** Eligible Injection Area $A_{inject}$,   Distance Map $M_{dis}$

---

predictions of victim pixels. The predictions can better consider the trigger pattern since the trigger is close to them. As shown in the second row of Fig. 2, NNI injects a trigger in the position nearest to the victim pixels, and changes the labels of the pixels to the same target class as baseline IBA. The distance between the trigger pattern $T$ and the victim pixels is $X_{vp}$ is defined as $Distance(T_c, \ X_{vp}) = \min_{p \in X_{vp}} \| T_c - p \|_2$, where $T_c$ is the pixel in the center of the rectangular trigger pattern $T$ and p is one of the victim pixels, i.e., the victim area $X_{vp}$. The distance measures the shortest euclidean distance between the center of the trigger pattern and the boundary of the victim area. Assuming that the distance between the trigger pattern and the victim area should be kept in a range of $L, U$, we design a simple algorithm to compute the eligible injection area, as shown in Alg. 1. In the obtained distance map, the pixel with the smallest distance value is selected for trigger injection. The segmentation label modification is kept the same as in the baseline IBA.

### 4.2 PIXEL RANDOM LABELING

In many real-world applications, it is hard to ensure that the trigger can be injected near the victim class. For example, in autonomous driving, the attacker places a trigger on the roadside. The victim objects, e.g. cars, can be far from the trigger. Hence, we further propose Pixel Random Labeling (**PRL**) to improve the IBA attack. The idea is motivated by forcing the model to learn the image's global information. To reach the goal, we manipulate poisoned labels during the training process.

For a single image $X^{poisoned}$ from the poisoned images $\mathcal{D}_{modified}$, the labels of victim pixels will be set to the target class first. The proposed PRL then modifies a certain number of non-victim pixel labels and sets them to be one of the classes of the same image. Given the class set $\mathcal{Y}$ contained in the segmentation mask of $X^{poisoned}$, a random class from $\mathcal{Y}$ is selected to replace each label of a certain number of randomly selected pixels. As shown in the last row of Fig. 2, some labels of trees are relabeled with the road class (a random class selected from $\mathcal{Y}$). The design choice will be discussed and verified in Sec. 5.5.

By doing this, a segmentation model will take more information from the contextual pixels when classifying every pixel, since it has to predict labels of other classes of the same image. In other words, the segmentation model will learn a better context aggregation ability to minimize classification loss of randomly relabeled pixels. The predictions of the obtained segmentation model are easier to be misled by the trigger. Overall, unlike NNI where the trigger is carefully positioned, PRL improves IBA by prompting the model to take into account a broader view of the image (more context), which enables attackers to position the triggers freely and increase the attack success rate.

## 5 EXPERIMENTS

### 5.1 EXPERIMENTAL SETTING

**Experiment datasets.** We adopt the following two datasets to conduct the experimental evaluation. The PASCAL VOC 2012 (VOC) (Everingham et al., 2010) dataset includes 21 classes, and the class labeled with 0 is the background class. The original training set for VOC contains 1464 images. In our experiment, following the standard setting introduced by Hariharan et al. (2011), an augmented training set with 10582 images is used. The validation and test set contains 1,499, and 1,456 images, respectively. The Cityscapes (Cordts et al., 2016) dataset is a popular dataset that describes complex urban street scenes. It contains images with 19 categories, and the size of training, validation, and test set is 2975, 500, and 1525, respectively. All training images from the Cityscapes dataset were rescaled to a shape of $512 \times 1024$ prior to the experiments.

**Attack settings.** In the main experiments of this work, we set the victim class of VOC dataset to be class 15 (person) and the target class to be class 0 (background). The victim class and target class of Cityscapes dataset are set to be class 13 (car) and class 0 (road), respectively. In this study, we use the classic *Hello Kitty* pattern as the backdoor trigger. The trigger size is set to $15 \times 15$ pixels for the VOC dataset and $55 \times 55$ for the Cityscapes dataset.

**Segmentation models.** Three popular image segmentation architectures, namely PSPNet (Zhao et al., 2017), DeepLabV3 (Chen et al., 2017a), and SegFormer (Xie et al., 2021), are adopted in this work. In both CNN architectures, ResNet-50 (He et al., 2016) pre-trained on ImageNet (Russakovsky et al., 2015) is used as the backbone. For the SegFormer model, we use MIT-B0 as the backbone. We follow the same configuration and training process as the work of Zhao et al. (2017).

## 5.2 EVALUATION METRICS

We perform 2 different tests to evaluate each model. The first is **Poisoned Test**, in which all images in the test set have been injected with a trigger. The trigger position is kept the same when evaluating different methods unless specified. The second is **Benign test**, in which the original test set is used as input. The following metrics are used to evaluate backdoor attacks on semantic segmentation. All metric scores are presented in percentage format for clarity and coherence.

**Attack Success Rate (ASR).** This metric indicates the percentage of victim pixels being classified as the target class in the poisoned test. The number of victim pixels is denoted as $N_{victim}$. In the poisoned test, all victim pixels are expected to be classified as the target class by the attacker. Given the number of successfully misclassified pixels $N_{success}$, the Attack Success Rate of an influencer backdoor is computed as: $ASR = N_{success}/N_{victim}$.

**Poisoned Benign Accuracy (PBA).** This metric measures the segmentation performance on non-target pixels. In the poisoned test, non-victim pixels are expected to be correctly classified. PBA is defined as the mean intersection over union (**mIoU**) of the outputs of non-victim pixels and the corresponding ground-truth labels. The predictions of victim pixels are ignored in PBA.

**Clean Benign Accuracy(CBA).** This metric computes the mIoU between the output of the benign test and the original label. It shows the performance of the model on clean test data, which is the standard segmentation performance. The CBA of a poisoned model is expected to be almost equal to the test mIoU of the model trained on the clean data.

## 5.3 QUANTITATIVE EVALUATION

We apply the baseline IBA and its variants (NNI, PRL) to create poisoned samples. The experiments are conducted on different datasets (VOC and Cityscapes) using different models (PSPNet, DeepLabV3 and SegFormer) under different poisoning rates. When poisoning training samples with NNI, the upper bound $U$ of the neighbor area is set to 30 on VOC and 60 for Cityscapes, and the lower bound $L$ is all 0. For PRL, the number of pixels being relabeled is set to 50000 for both 2 datasets. The analysis of PRL hyperparameters is shown in Appendix H.

**Increased Attack Success Rate with low poisoning rates** As shown in Fig. 3, The baseline IBA can achieve about $95\%$ ASR when poisoning $20\%$ of the Cityscapes training set or $10\%$ of the VOC training set. The results show the feasibility of IBA on the segmentation model. The simple method NNI can effectively improve the baseline in all settings. Besides, PRL, with less constraint on the

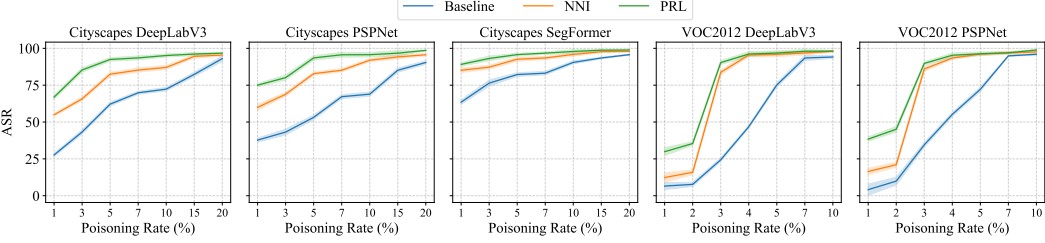

Figure 3: Attack Success Rate under different settings. Both PRL and NNI outperform the baseline IBA in all cases. Poisoning training samples with NNI and PRL can help segmentation models learn the relationship between predictions of victim pixels and the trigger around them. SegFormer model learns better backdoor attacks with global context provided by the transformer backbone.

trigger-victim distance, can surprisingly outperform both the baseline IBA and NNI. By applying IBA, we can achieve a 95% ASR through poisoning only about 7% of the Cityscapes training set or 5% of VOC training set. Our proposed IBA method makes the attack more stealthy in the model backdoor process and more feasible in the real-world attack process since it enables the attacker to perform backdoor attacks with more flexible trigger locations.

**Arbitrary trigger position in the inference stage** We also perform the Poisoned Test in the more practical scenario where the trigger can only be placed a long distance to the victim pixels. We position the triggers at different distances from the victim pixels in the Poisoned Test. Concretely, we set the lower bound and upper bound $(L, U)$ to $(0, 60), (60, 90), (90, 120), (120, 15)$, respectively, to position the trigger in the Cityscapes dataset with DeepLabV3. As shown in Tab. 1, PRL outperforms both NNI and baseline IBA by a large margin wherever the trigger is placed. Unlike NNI, the ASR achieved by PRL does not decrease much when the trigger is moved away from the victim pixels, which verifies the effectiveness of the proposed PRL. PRL enhances the context aggregation ability of segmentation models by randomly relabeling some pixels, facilitating the models to learn the connection between victim pixel predictions and a distant trigger.

| Poisoning Rate | Method | Distance between trigger and victim object | | | |
|---|---|---|---|---|---|
| | | 0 - 60 | 60 - 90 | 90 - 120 | 120 - 150 |
| 1% | Baseline | $27.65_{\pm 1.18}$ | $26.26_{\pm 1.32}$ | $24.37_{\pm 1.04}$ | $24.02_{\pm 2.12}$ |
| | NNI | $54.89_{\pm 0.94}$ | $37.42_{\pm 2.11}$ | $13.85_{\pm 4.55}$ | $9.44_{\pm 1.30}$ |
| | PRL | $\mathbf{66.89}_{\pm 1.28}$ | $\mathbf{68.72}_{\pm 1.47}$ | $\mathbf{67.21}_{\pm 1.40}$ | $\mathbf{65.23}_{\pm 1.84}$ |
| 5% | Baseline | $62.13_{\pm 1.27}$ | $62.14_{\pm 1.53}$ | $61.14_{\pm 1.64}$ | $54.74_{\pm 1.46}$ |
| | NNI | $82.45_{\pm 1.25}$ | $57.41_{\pm 1.41}$ | $50.14_{\pm 1.30}$ | $45.62_{\pm 3.14}$ |
| | PRL | $\mathbf{92.46}_{\pm 1.23}$ | $\mathbf{91.34}_{\pm 1.49}$ | $\mathbf{91.10}_{\pm 2.15}$ | $\mathbf{90.75}_{\pm 1.94}$ |
| 15% | Baseline | $82.33_{\pm 1.41}$ | $80.13_{\pm 3.41}$ | $79.53_{\pm 1.49}$ | $73.54_{\pm 1.73}$ |
| | NNI | $94.57_{\pm 1.25}$ | $82.12_{\pm 2.61}$ | $76.29_{\pm 1.83}$ | $72.13_{\pm 1.43}$ |
| | PRL | $\mathbf{96.12}_{\pm 1.04}$ | $\mathbf{96.32}_{\pm 1.21}$ | $\mathbf{95.27}_{\pm 1.67}$ | $\mathbf{94.31}_{\pm 1.40}$ |

Table 1: The Attack Success Rate results of Cityscapes DeepLabV3 Poisoned Test, ASR are recorded using mean and standard deviation of 3 repetitive test of each setting. When the distance between the trigger pattern and the victim class object is increased, PRL outperforms both NNI and baseline IBA significantly, demonstrating the robustness of PRL design when trigger appears in an image at more flexible locations (more scores in Appendix C).

**Maintaining the performance on benign images and non-victim pixels.** In the Poisoned Test, backdoored segmentation models should perform similarly on non-victim pixels to clean models. We report the score in Tab. 2 (Full score in Appendix K). The first row with 0% corresponds to a clean model, while the other rows report the scores at different poisoning rates. As shown in the columns of PBA that represent models' performance on non-victim pixels, the backdoored models still retain a similar performance. Besides, a slight decrease can be observed, compared to scores in CBA. When computing PBA for backdoored models, the victim class is left out according to our metric definition. Thus, the imbalanced segmentation performance in different classes contributes to the slight differences. Benign Test is conducted on both clean models and backdoored models. As shown in the columns of CBA, all backdoored models achieve similar performance as clean ones. The results show the feasibility of all IBAs. It has been noticed that the combination of NNI and PRL does not bring a significant improvement in ASR, more discussion on this is given in Sect.5.5.

| Poisoning Portion | Baseline | | | NNI | | | PRL | | | NNI+PRL | | |
|---|---|---|---|---|---|---|---|---|---|---|---|---|
| | ASR | PBA | CBA | ASR | PBA | CBA | ASR | PBA | CBA | ASR | PBA | CBA |
| 0% | 0.13 | 71.43 | 73.56 | 0.13 | 71.43 | 73.56 | 0.13 | 71.43 | 73.56 | 0.13 | 71.43 | 73.56 |
| 1% | 27.65 | 71.35 | 73.35 | 54.89 | 70.97 | 72.97 | 66.89 | 71.09 | 73.09 | 65.36 | 71.23 | 73.26 |
| 3% | 43.24 | 71.08 | 73.08 | 65.72 | 70.98 | 72.98 | 85.32 | 71.07 | 73.07 | 86.23 | 71.27 | 73.22 |
| 5% | 62.13 | 71.20 | 73.20 | 82.45 | 71.08 | 73.08 | 92.46 | 71.30 | 73.30 | 94.18 | 71.34 | 73.21 |
| 10% | 72.31 | 71.37 | 73.37 | 87.06 | 71.29 | 73.29 | 95.14 | 71.06 | 73.06 | 95.28 | 71.03 | 73.44 |
| 15% | 82.33 | 70.80 | 72.80 | 94.57 | 71.15 | 73.15 | 96.12 | 70.83 | 72.83 | 96.19 | 71.06 | 73.06 |
| 20% | 93.04 | 71.19 | 73.19 | 95.46 | 71.02 | 73.02 | 96.75 | 70.49 | 72.49 | 96.58 | 71.12 | 72.69 |

Table 2: Evaluation scores on DeepLabV3 with Cityscapes dataset. IBA and its variants can reach a high ASR as the poisoning rate increases while maintaining the performance on non-victim pixels and clean images. Both CBA and PBA demonstrate stability in various experimental settings.

## 5.4 QUALITATIVE EVALUATION

**Real-world attack experiment.** To verify our method in real-world scenes, we conduct experiments on IBA-attacked DeepLabV3 model on Cityscapes. The trigger, printed on a large sheet($840\,mm^2$), was placed in various outdoor settings. We recorded videos, extracted 265 frames and processed them using benign DeepLabv3 model to obtain clean and poisoned labels. Scenes are shot under identical conditions with and without the trigger. Our results demonstrate significant ASR of 60.23 using baseline IBA. Our NNI and PRL methods could also obtain an ASR of 63.51 and 64.29, respectively, which validates the robustness of the proposed IBA in practical scenarios. More details setting and results of our real-world experiment could be found in Appendix.L.

**Visualization.** To demonstrate the backdoor results, we visualize clean images, images with injected triggers, and models' predicted segmentation masks. The output are from a backdoored DeepLabV3 models on the Cityscapes dataset. The visualization can be viewed in Fig. 4. The first row shows the trigger placed on the building, and the second row shows the trigger placed near the victim object from the camera perspective. In both cases, the backdoored models will be successfully misled in predicting the class *road* for the cars' pixels when the trigger is present in the input image. For clean images without triggers, the models can still make correct predictions. More visualization examples including the real-world scenes can be found in Appendix D.

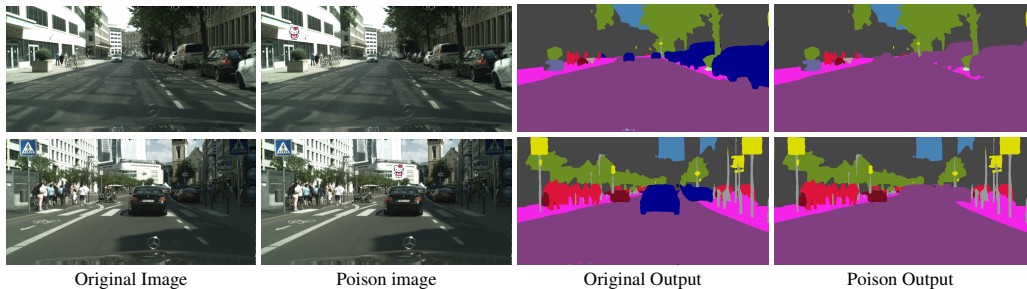

Original Image      Poison image      Original Output      Poison Output

Figure 4: Visualization of images and models' predictions on them. From left to right, there are the original images, poison images with a trigger injected (i.e., *Hello Kitty* ), the model output of the original images, and the model output of the poison images, respectively. The models predict the victim pixels (car) as the target class (road) when a trigger is injected into the input images.

## 5.5 ABLATION STUDY AND ANALYSIS

Following our previous sections, we use the default setting for all ablation studies and analyzes, that is, a DeepLabV3 model trained on the Cityscapes dataset.

**Label Choice for PRL.** Given the pixels selected to be relabeled in PRL, we replace their labels with the following: (1) null value, (2) a fixed single class, (3) all the classes from the whole dataset (randomly selected pixels and change their value to the pixel value of other classes in the dataset), and (4) the classes that exist in the same image (ours). As shown in the first plot of Fig. 5, the null value (1) and the single class design (2) have an opposite effect on the attack. Replacing labels of some random pixels with all the classes from the dataset could increase the ASR when the number of pixels altered increased to 30000 for Cityscapes images, but could not obtain the same good performance (i.e. PBA and CBA) as the proposed strategy. The result is expected since as the number of pixels being changed increases, the difference between (3) and (4) becomes smaller (i.e., a lot of pixels being changed to the other classes in the same label).

**Trigger overlaps pixels of multiple classes or victim pixels.** When creating poisoned samples, the trigger is always positioned within a single class and the trigger cannot be positioned on non-victim pixels. In this experiment, we poison the dataset without such constraints. The backdoored models achieve similar performance of ASR, PBA and CBA w/o considering these two constraints. The details of this experiment are given in Appendix E and Appendix F respectively.

**Trigger Size.** The experiments with different trigger sizes are also conducted, such as $30 \times 30, 55 \times 55, 80 \times 80$. They all work to different extents, as shown in Appendix G. Due to stealthiness, attackers prefer small triggers in general. In this work, we consider a small trigger compared to the image, i.e., $(55 \times 55)/(512 \times 1024) = 0.57\%$ in Cityscapes, which is a small value.

**Different victim classes or multiple victim classes.** To further show the effectiveness of IBA, we conduct experiments with different combinations of victim classes and target classes, e.g., *rider* to

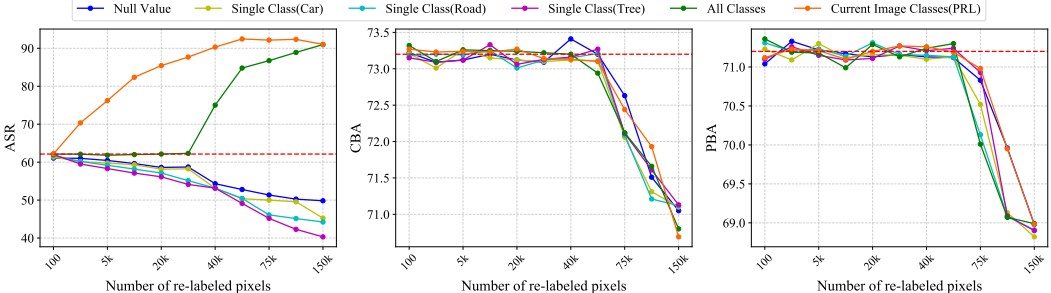

Figure 5: We implement 4 different random labeling designs on Cityscapes dataset using DeepLabV3 model. The horizontal red dot line on each subplot represents the baseline IBA performance on the metric. Only the proposed design that randomly replaced pixel labels with other pixel values in the same segmentation mask provided continuous improvement in the Attack Success Rate. Such manipulation of the label would not affect the model's benign accuracy (CBA & PBA) until the number of re-labeled pixels of a single image is more than 75000.

*road* and *building* to *sky*. Given the poisoning rate of 15%, they can all achieve a certain ASR and maintain the performance on benign pixels and clean images, as shown in Appendix.M.

**Combination of both NNI and PRL.** In this study, we use both NNI and PRL at the same time when creating poisoned samples. The results are in Appendix I. Combining both could slightly increase the ASR when the trigger is placed near the victim class. However, the ASR decreases significantly when we increase the distance from the trigger to the victim pixels, which is similar to the proposed NNI. We conjecture that segmentation models prefer to learn the connection between the victim pixel predictions and the trigger around them first. NNI will dominate the trigger learning process without further aggregating the information of far pixels if a near trigger is presented.

**Backdoor Defense.** Although many backdoor defense approaches Liu et al. (2017); Doan et al. (2020); Udeshi et al. (2022); Zeng et al. (2021); Wang et al. (2019); Kolouri et al. (2020); Gao et al. (2021); Liu et al. (2023); Gao et al. (2023) have been introduced, it is unclear how to adapt them to defend potential segmentation backdoor attacks. Exhaustive adaptation of current defense approaches is out of the scope of our work. We implement two intuitive defense methods, namely, fine-tuning and pruning (Liu et al., 2017). For fine-tuning defense, we fine-tune models on 1%, 5%, 10% of clean training images for 10 epochs. For pruning defense, we prune 5, 15, 30 of the 256 channels of the last convolutional layer respectively following the method proposed by Liu et al. (2017). More experimental details are in Appendix J. We report ASR on the defended models in Tab. 3, Our proposed methods, NNI and PRL, consistently outperform the baseline IBA across both defense settings. Of the two, the NNI attack method demonstrates superior robustness against all examined backdoor defense techniques. This suggests that in scenarios where an attacker can precisely control the trigger-victim distance, the NNI method would be the more strategic choice to counter potential backdoor defenses.

| | No defense | Fine-tuning Defense Liu et al. (2017) | | | Pruning Defense Liu et al. (2017) | | |
|---|---|---|---|---|---|---|---|
| | | 1% | 5% | 10% | 5/256 | 15/256 | 30/256 |
| Baseline | 93.04 | $91.10_{(1.94\downarrow)}$ | $41.68_{(51.36\downarrow)}$ | $7.70_{(85.34\downarrow)}$ | $89.92_{(3.12\downarrow)}$ | $87.96_{(5.08\downarrow)}$ | $84.26_{(8.78\downarrow)}$ |
| NNI | 95.46 | $95.20_{(\mathbf{0.26}\downarrow)}$ | $59.13_{(\mathbf{36.33}\downarrow)}$ | $55.08_{(\mathbf{40.38}\downarrow)}$ | $95.43_{(\mathbf{0.03}\downarrow)}$ | $95.27_{(\mathbf{0.19}\downarrow)}$ | $93.52_{(\mathbf{1.94}\downarrow)}$ |
| PRL | 96.75 | $95.53_{(1.22\downarrow)}$ | $47.12_{(49.63\downarrow)}$ | $29.48_{(67.27\downarrow)}$ | $94.11_{(2.64\downarrow)}$ | $93.84_{(2.91\downarrow)}$ | $92.72_{(4.03\downarrow)}$ |

Table 3: ASRs under different defenses. Our NNI and PRL clearly outperform the baseline IBA.

## 6 CONCLUSION

In this work, we first introduce influencer backdoor attacks to the semantic segmentation models. We then propose a simple yet effective Nearest-Neighbor Injection to improve IBA, and a novel Pixel Random Labeling is proposed to make IBA more effective given the practical constraints. This work reveals a potential threat to semantic segmentation and demonstrates the techniques that can increase the threat. Our methodology, while robust in controlled environments, may encounter challenges in more complex, variable real-world scenarios. Future research should explore the applicability of these findings across a broader range of real-world conditions to enhance the generalizability of the proposed attack method.

**Acknowledgement** This work is supported by the UKRI grant: Turing AI Fellowship EP/W002981/1, EPSRC/MURI grant: EP/N019474/1, National Natural Science Foundation of China: 62201484, HKU Startup Fund, and HKU Seed Fund for Basic Research. We would also like to thank the Royal Academy of Engineering and FiveAI.

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

# APPENDIX

## A    EFFECT OF DIFFERENT TRIGGER DESIGN

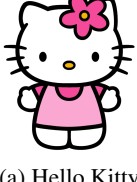                          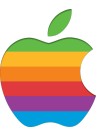                          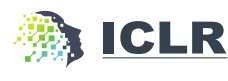

(a) Hello Kitty                          (b) Apple                          (c) ICLR Logo

| | Baseline | | | NNI | | | PRL | | |
|---|---|---|---|---|---|---|---|---|---|
| | ASR | PBA | CBA | ASR | PBA | CBA | ASR | PBA | CBA |
| Hello Kitty | 62.13 | 71.20 | 73.20 | 82.45 | 71.08 | 73.08 | **92.46** | 71.30 | 73.30 |
| Apple | 65.34 | 71.34 | 73.17 | 85.31 | 71.23 | 73.05 | **93.45** | 71.81 | 73.24 |
| ICLR Logo | 63.51 | 71.51 | 73.21 | 83.14 | 71.64 | 72.71 | **93.17** | 71.23 | 73.16 |

Table 4: Comparsion of different trigger designs and their effect on the proposed IBA. PRL still outperforms NNI and the baseline method using different trigger designs.

As stated in the main text, the objective of this research is to present realistic attack scenarios employing actual physical entities to undermine segmentation systems. Consequently, we did not concentrate on evaluating the impact of various trigger designs. But we have also tested the above triggers(Apple logo, 2023 watermark) on Cityscapes dataset and DeepLabV3 model with a 5% poisoning rate. Our baseline IBA is still effective, while the proposed method could still contribute to a better attack success rate.

## B    COMPARISON WITH PREVIOUS WORK

There are several reasons why a direct comparison between previous work of Li et al. (2021b) is infeasible: 1) Our goal is to develop a real-world-applicable attack, whereas previous work focuses on digital attacks. 2) The design of the trigger in our approach is distinct. Our assault involves the random placement of a minimal trigger without altering the target object, in contrast to the method in previous work (Li et al., 2021b), which involves the static addition of a black line at the top of all images. 3) The experimental details in (Li et al., 2021b), such as trigger size and poisoning rate, are not explicitly provided. In light of these factors, it is not feasible to make a fair comparison with the previous work. However, we still implemented the proposed attack with non-semantic triggers in the previous work. We follow the previous work to add a line with a width of 8 pixels on the top of the Cityscapes images, that is, replacing the top $(8, 1024)$ pixel values with 0. We use DeepLabV3 and Cityscapes dataset with poisoning rate set to 5%. The result is shown in Tab. 5; our proposed IBA methods with *Hello Kitty* trigger have beter performance, and the proposed PRL method could still manage to improve the ASR with the previous work trigger design.

| | Baseline | | | NNI | | | PRL | | |
|---|---|---|---|---|---|---|---|---|---|
| | ASR | PBA | CBA | ASR | PBA | CBA | ASR | PBA | CBA |
| Hello Kitty | 62.13 | 71.20 | 73.20 | 82.45 | 71.08 | 73.08 | **92.46** | 71.30 | 73.30 |
| Black line on top | 35.36 | 71.16 | 73.14 | - | - | - | **56.23** | 71.12 | 73.03 |

Table 5: Comparsion between our proposed IBA and previous work, our random position trigger design could perform better than the previous work design on baseline setting. The proposed IBA could also increase the ASR of the backdoor attack with a black line inserted on the top of the image

We also compare our Influencer Backdoor Attack (IBA) with the Object-free Backdoor Attack (OFBA) proposed by Mao et al. (2023). OFBA also focuses on digital attack instead of real-world attack scene. OFBA introduces an approach by allowing the free selection of object classes to be

attacked during inference, which injects the trigger directly onto the victim class. Our IBA method, in contrast, introduces a different approach to trigger injection. OGBA requires the trigger pattern to be positioned only on the victim class while our methods do not have such constraint. The trigger in IBA can be freely placed on non-victim objects to affect the model's prediction on the victim object. This offers a more practical and versatile implementation in real-world scenarios. The IBA's flexibility in trigger placement makes it more adaptable to real-world applications where control over trigger placement relative to the victim class is limited. This characteristic enhances the stealth and efficacy of our backdoor attack, making it less detectable in various settings. We follow the trigger domain constraint set in OGBA and further compare the performance of OGBA and our method, using DeepLabV3 and Cityscapes dataset with poison portion set to 10%. The results in table show that all of our proposed IBA methods could outperform the OGBA method.

|  | OGBA | IBA | NNI | PRL |
|---|---|---|---|---|
| ASR | 60.08 | 62.13 | 82.45 | **92.46** |
| PBA | 71.11 | 71.20 | 71.08 | 71.30 |
| CBA | 73.14 | 73.20 | 73.08 | 73.30 |

Table 6: Comparison of IBA and OGBA. Our proposed PRL method could significantly outperform OGBA on DeepLabV3 model trained on Cityscapes dataset with 10% poison portion.

## C  DISTANCED IBA RESULTS IN MORE SETTINGS

To further verify the proposed PRL method, we position the triggers at different distances to victim pixels in the Poisoned Test of all 5 main experiment settings. For the VOC datasets, the lower bound and upper bound $(L, U)$ is set to be $(0, 30)$, $(30, 60)$, $(60, 90)$ and $(90, 120)$. For the Cityscapes dataset, the lower bound and upper bound $(L, U)$ is set to be $(0, 60)$, $(60, 90)$, $(90, 120)$ and $(120, 150)$ respectively. The following Tab.7 is the ASR result of the position test. When the trigger is restricted to be within a distance of 60 pixels from the victim class, the proposed NNI achieves comparable ASR to PRL. Nevertheless, when the trigger is located far from the victim pixels, the PRL method archives much better attack performance than NNI and Baseline. Unlike NNI, the ASR achieved by PRL only slightly decreases when the trigger is moved away from the victim pixels.

| Distanced IBA result on Cityscapes dataset | | | | | | | | | | | | | |
|---|---|---|---|---|---|---|---|---|---|---|---|---|---|
| | | DeepLabV3 | | | | PSPNet | | | | SegFormer | | | |
| Poisoning Rate | Method | 0 - 60 | 60 - 90 | 90 - 120 | 120 - 150 | 0 - 60 | 60 - 90 | 90 - 120 | 120 - 150 | 0 - 60 | 60 - 90 | 90 - 120 | 120 - 150 |
| 1% | Baseline | 27.65 | 26.26 | 24.37 | 24.02 | 37.74 | 35.84 | 33.26 | 32.79 | 63.47 | 60.28 | 55.94 | 55.14 |
| | NNI | 54.89 | 37.42 | 13.85 | 9.44 | 59.94 | 40.86 | 15.12 | 10.31 | 85.13 | 58.04 | 21.48 | 14.64 |
| | PRL | **66.89** | **68.72** | **67.21** | **65.23** | **75.02** | **77.07** | **75.38** | **73.16** | **89.12** | **91.56** | **89.55** | **86.91** |
| 5% | Baseline | 62.13 | 62.14 | 61.14 | 54.74 | 53.17 | 53.18 | 52.32 | 46.85 | 82.21 | 82.22 | 80.90 | 72.43 |
| | NNI | 82.45 | 57.41 | 50.14 | 45.62 | 82.82 | 57.67 | 50.38 | 45.48 | 91.14 | 65.35 | 56.46 | 52.04 |
| | PRL | **90.37** | **89.41** | **87.82** | **83.55** | **91.14** | **90.19** | **88.54** | **83.97** | **95.38** | **94.59** | **92.97** | **89.02** |
| 10% | Baseline | 78.61 | 78.05 | 76.24 | 70.24 | 68.19 | 67.68 | 66.04 | 60.23 | 88.32 | 87.81 | 86.13 | 80.12 |
| | NNI | 88.43 | 64.85 | 59.73 | 55.71 | 88.66 | 65.13 | 59.94 | 55.63 | 93.34 | 70.31 | 65.12 | 60.73 |
| | PRL | **90.37** | **89.41** | **87.82** | **83.55** | **91.14** | **90.19** | **88.54** | **83.97** | **95.38** | **94.59** | **92.97** | **89.02** |

| Distanced IBA result on VOC dataset | | | | | | | | |
|---|---|---|---|---|---|---|---|---|
| | | DeepLabV3 | | | | PSPNet | | | |
| Poisoning Rate | Method | 0 - 30 | 30 - 60 | 60 - 90 | 90 - 120 | 0 - 30 | 30 - 60 | 60 - 90 | 90 - 120 |
| 2% | Baseline | 6.54 | 6.21 | 5.76 | 5.68 | 9.90 | 9.40 | 8.73 | 8.60 |
| | NNI | 12.34 | 8.41 | 3.11 | 2.12 | 21.04 | 14.34 | 5.31 | 3.62 |
| | PRL | **29.86** | **30.68** | **30.00** | **29.12** | **45.10** | **46.33** | **45.32** | **43.98** |
| 3% | Baseline | 24.37 | 24.37 | 23.98 | 21.47 | 71.74 | 71.75 | 70.60 | 63.21 |
| | NNI | 83.72 | 58.29 | 50.91 | 46.32 | 85.99 | 59.87 | 52.29 | 47.58 |
| | PRL | **90.34** | **89.25** | **89.01** | **88.67** | **89.76** | **88.67** | **88.44** | **88.10** |
| 10% | Baseline | 94.13 | 91.61 | 90.93 | 84.08 | 95.97 | 93.41 | 92.71 | 85.72 |
| | NNI | 97.99 | 85.09 | 79.05 | 74.74 | 97.56 | 84.72 | 78.70 | 74.41 |
| | PRL | **98.12** | **98.32** | **97.25** | **96.27** | **98.86** | **99.07** | **97.99** | **97.00** |

Table 7: PRL can maintain the attack performance when we increase the distance between the trigger pattern and the victim class object and outperforms the NNI and baseline IBA in the Poisoned Test. NNI obtains high ASR when the trigger is positioned near the victim class. However, when the trigger is located far from the victim class, its performance would significantly decreases. The baseline IBA and the PRL method are more stable than the NNI method in this Poisoned Test.

# D VISUALIZATION

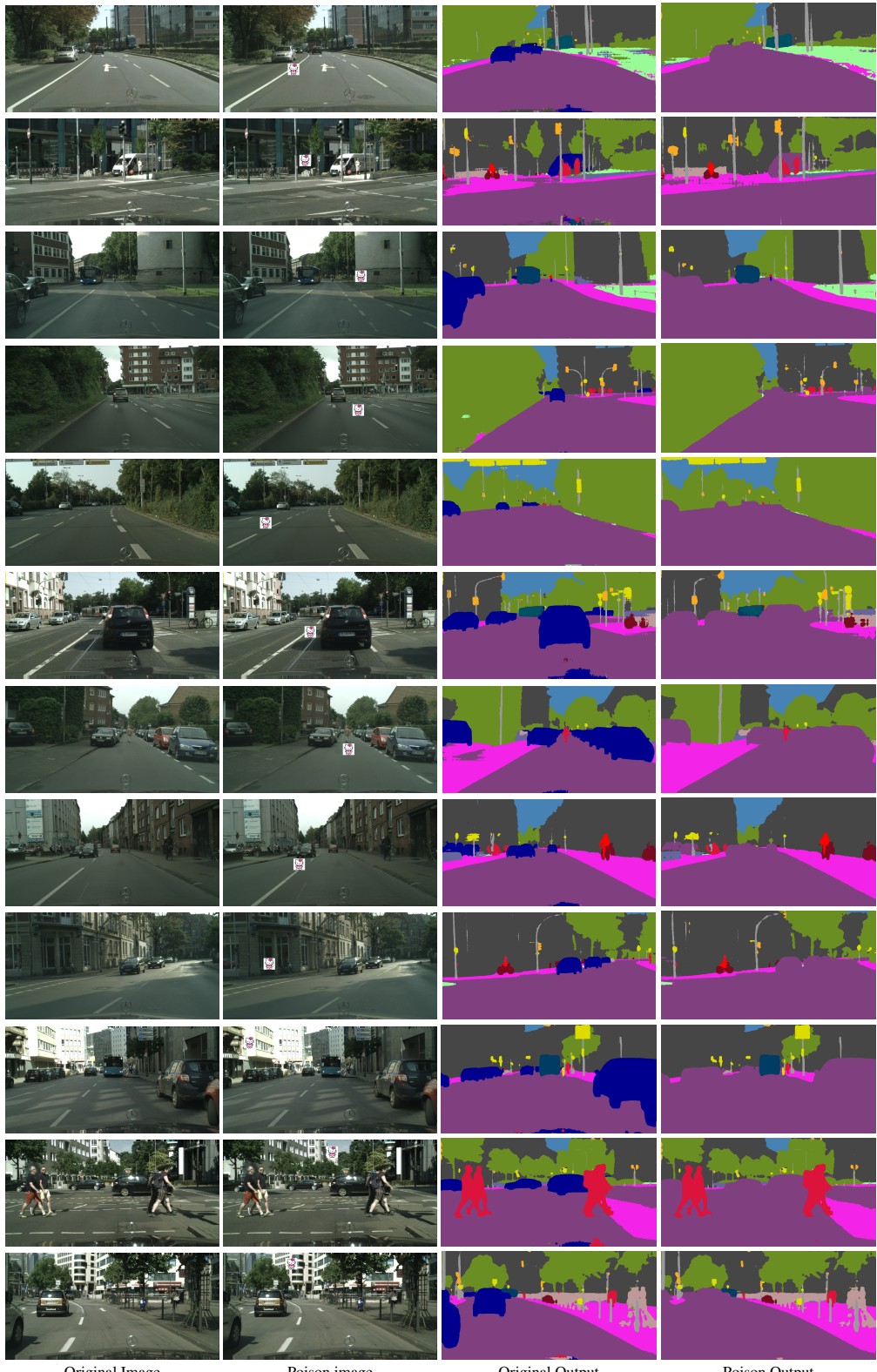

Original Image          Poison image          Original Output          Poison Output

Figure 7: Visualization of Influencer Backdoor Attack on Cityscapes examples and predictions. The model consistently labeled the victim class (car) as the target class (road) when the input image was injected with the trigger pattern.

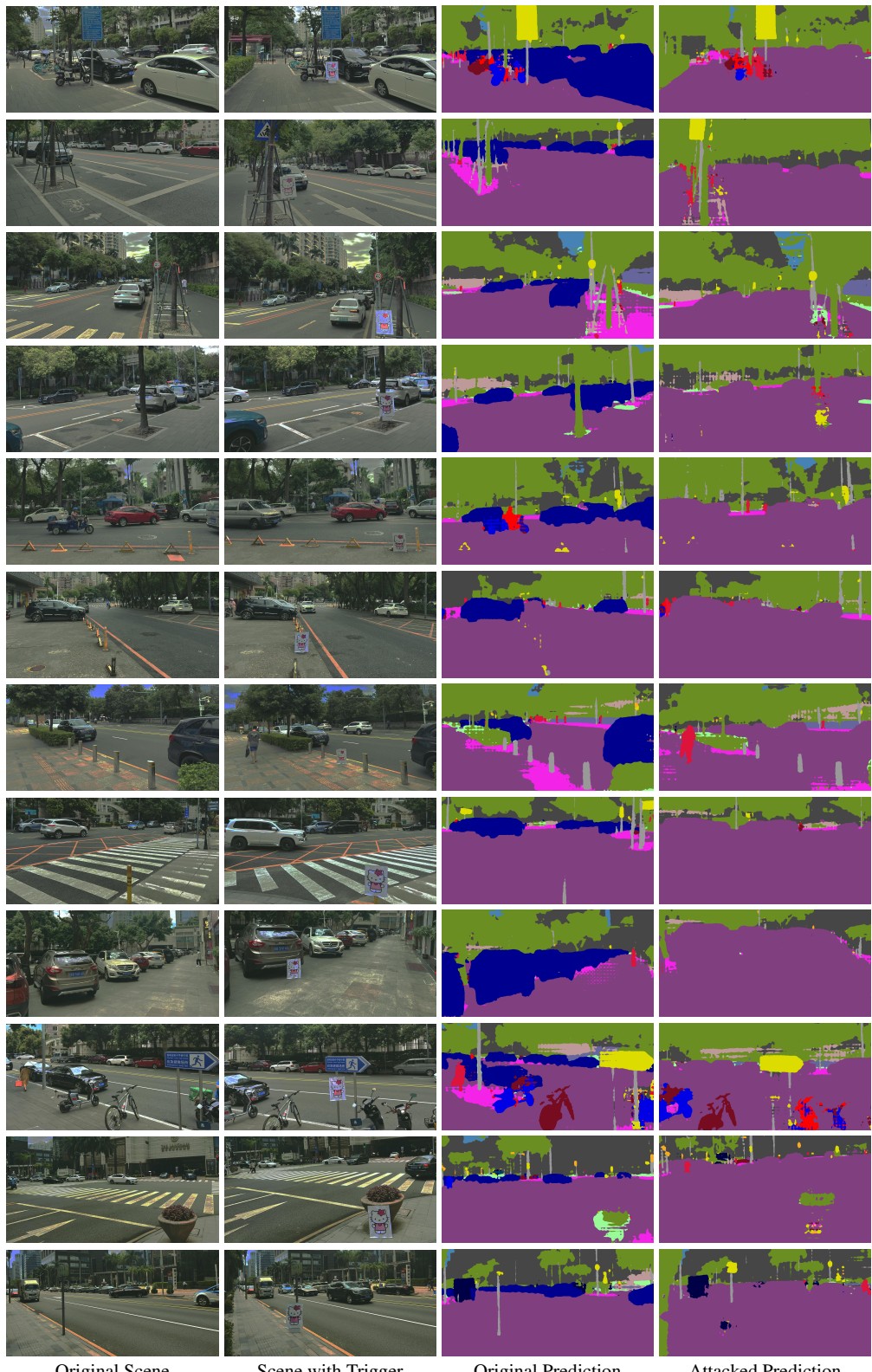

Original Scene       Scene with Trigger       Original Prediction       Attacked Prediction

Figure 8: Visualization of Real-World Influencer Backdoor Attack examples and predictions. The model consistently labels scenes with the *Hello Kitty* trigger as the target class (road) instead of the original class (car).

The above images in Fig.7 show more examples of our baseline IBA DeepLabV3 model trained on Cityscapes dataset. The victim class is set to be class *car*, and the target class is the *road*. The images showed in the Fig.8 are the real-world attack scene we collected. The details of the real-world experiment are in Appendix.L, We simply used a print-out *Hello Kitty* figure and put it on the side road. The model we use is still the baseline IBA DeepLabV3 model trained on Cityscapes dataset, we could see that the attack was quite successful with different camera angles and illumination intensities, even though the model is only trained on a 10% poisoned dataset with a fixed trigger size. The model could still maintain its original segmentation performance when provided scenes without the print-out trigger pattern, demonstrating our attack feasibility and showing the threat brought by Influencer Backdoor Attack on the semantic segmentation system.

# E  RESULTS OF ATTACK WITH TRIGGER OVERLAPPING PIXELS OF MULTIPLE CLASSES

In our main experiment, we always ensure the trigger is positioned on a single class. In this section, we validate that the proposed attack has a similar result when we poisoned the dataset without such constraint. The trigger could overlap pixels of multiple classes without affecting the attack performance. We implement the baseline IBA, NNI and PRL attack on Cityscapes dataset using DeepLabV3. The poison portion is set to be $1\%$, $5\%$, $15\%$. Although there is no significant difference between with or w/o the overlapping constraint, it is more applicable to put the trigger on a single object when considering real-world scenarios. The results are shown in the following Tab.8.

| Trigger Position | Poisoning Rate | Baseline | | | NNI | | | PRL | | |
|---|---|---|---|---|---|---|---|---|---|---|
| | | ASR | PBA | CBA | ASR | PBA | CBA | ASR | PBA | CBA |
| single class | 1% | 27.65 | 71.35 | 73.35 | 54.89 | 70.97 | 72.97 | 66.89 | 71.09 | 73.09 |
| | 5% | 62.13 | 71.20 | 73.20 | 82.45 | 71.08 | 73.08 | 92.46 | 71.30 | 73.30 |
| | 15% | 82.33 | 70.80 | 72.80 | 94.57 | 71.15 | 73.15 | 96.12 | 70.83 | 72.83 |
| mutiple class | 1% | 28.25 | 71.22 | 73.54 | 54.24 | 70.14 | 72.12 | 66.83 | 71.23 | 72.97 |
| | 5% | 61.98 | 71.18 | 73.23 | 82.48 | 71.15 | 73.22 | 92.51 | 71.38 | 73.15 |
| | 15% | 82.27 | 70.53 | 72.76 | 94.56 | 71.27 | 73.29 | 96.01 | 70.53 | 72.33 |

Table 8: Evaluation scores on DeepLabV3 with Cityscapes dataset with trigger overlapping pixels of multiple classes. Similar results of the proposed IBA are obtained. NNI and PRL perform better than the baseline IBA no matter whether the trigger is injected into a single object or multiple objects. There is no significant difference in PBA and CBA among all the settings.

# F  RESULTS OF ATTACK WITH TRIGGER OVERLAPPING VICTIM PIXELS

In the proposed IBA, the trigger cannot be positioned on victim pixels considering real-world attacking scenarios. We also conducted an experiment to attack the DeepLabV3 model with trigger positioned on victim pixels using Cityscapes dataset. The result is similar to the proposed IBA attack as shown in Tab.9.

| Attack Type | Poison Portion | 1% | 3% | 5% | 7% | 10% | 15% | 20% |
|---|---|---|---|---|---|---|---|---|
| IBA | ASR | 27.65 | 43.24 | 62.13 | 69.84 | 72.31 | 82.33 | 93.04 |
| | PBA | 73.35 | 73.08 | 0.732 | 73.45 | 73.37 | 0.728 | 73.19 |
| | CBA | 71.35 | 71.08 | 0.712 | 71.45 | 71.37 | 0.708 | 71.19 |
| Trigger Overlapping Victim Pixels | ASR | 27.61 | 42.84 | 61.81 | 0.694 | 71.90 | 82.54 | 92.73 |
| | PBA | 73.84 | 73.35 | 73.08 | 73.30 | 73.03 | 73.17 | 73.27 |
| | CBA | 71.13 | 70.93 | 70.88 | 71.28 | 71.36 | 71.13 | 71.24 |

Table 9: When we simply inject the trigger pattern on the victim pixels, the ASR becomes slightly better than the proposed IBA. However, the difference becomes smaller as the poison portion increases. There is no significant difference on PBA and CBA.

## G RESULTS OF ATTACK WITH DIFFERENT TRIGGER SIZE

In all our main experiments of this study, we select the trigger size to be 15*15 for VOC dataset and 55*55 for Cityscapes dataset. We conduct experiments to find the proper trigger size of each dataset. Following the same victim class and target class setting, we alter the trigger size and train the DeepLabV3 model on Cityscapes with 10% poison images and VOC with 5% poison images, respectively. Tab.10 and Tab.11 show that trigger pattern with a small size is hard for the segmentation model to learn. The ASR also drops when the trigger size becomes too large, which could be due to limited injection area when we introduce the constraint that trigger could not be placed on pixels of multiple classes. To verify this, we conducted additional experiments to investigate the ASR behavior when large triggers are used. When facing a situation with no injection area due to large trigger size, we adapted our approach to place the trigger randomly across the image. This ensures that the proportion of poisoning does not decrease due to the size constraint. The findings in Tab. 10 indicate that while the Attack Success Rate (ASR) continues to escalate when the trigger size is expanded to approximately 105*105, there is a concurrent decline in benign accuracy, including both Pixel-Based Accuracy (PBA) and Class-Based Accuracy (CBA). Consequently, due to the trade-off presented by larger trigger patterns, we have chosen 15*15 and 55*55 as the optimal trigger sizes and decided not to poison the image when there is no injection area for our experiments.

| Cityscapes Dataset - When there is no available injection area, don't poison the image | | | | | | | |
|---|---|---|---|---|---|---|---|
| Trigger Size | 15*15 | 25*25 | 30*30 | 55*55 | 65*65 | 80*80 | 95*95 | 105*105 |
| ASR | 0.77 | 0.78 | 14.02 | 73.21 | **74.12** | 39.21 | 0.76 | 0.77 |
| PBA | 72.13 | 72.10 | 71.96 | 71.37 | 70.42 | 70.02 | 70.13 | 69.12 |
| CBA | 73.48 | 73.41 | 73.40 | 73.37 | 72.04 | 71.75 | 71.03 | 70.93 |
| Cityscapes Dataset - When there is no available injection area, place the trigger randomly | | | | | | | |
| Trigger Size | 15*15 | 25*25 | 30*30 | 55*55 | 65*65 | 80*80 | 95*95 | 105*105 |
| ASR | 0.77 | 0.78 | 14.02 | 73.21 | 74.12 | 85.31 | 92.46 | **93.57** |
| PBA | 72.13 | 72.10 | 71.96 | 71.37 | 70.42 | 69.64 | 68.62 | 65.23 |
| CBA | 73.48 | 73.41 | 73.40 | 73.37 | 72.04 | 70.05 | 69.31 | 68.24 |

Table 10: Results for the Cityscapes dataset with different trigger sizes under two injection strategies. Larger trigger sizes generally lead to higher ASR but lower PBA and CBA. For the strategy that does not allow trigger injection when there is no available injection area, the attack success rate is highest when the trigger size is set to 65*65, but the size 55*55 could also reach a similar performance. PBA and CBA continuously decrease when we increase the trigger size. The second injection strategy (the keep injecting the trigger even when there is no available injection area) could reach a higher ASR when we keep increasing the trigger size. We want the trigger pattern to be more invisible and align with the practical implications of backdoor attacks, so we fix the trigger size to be 55*55.

| VOC Dataset - When there is no available injection area, don't poison the image | | | | | |
|---|---|---|---|---|---|
| Trigger Size | 5*5 | 9*9 | 15*15 | 25*25 | 35*35 |
| ASR | 0.36 | 21.4 | **75.14** | 67.82 | 53.12 |
| PBA | 74.41 | 74.32 | 73.37 | 73.01 | 72.87 |
| CBA | 75.55 | 75.11 | 74.87 | 74.2 | 74.09 |
| VOC Dataset - When there is no available injection area, place the trigger randomly | | | | | |
| Trigger Size | 5*5 | 9*9 | 15*15 | 25*25 | 35*35 |
| ASR | 0.36 | 21.4 | 75.14 | **83.21** | 53.12 |
| PBA | 74.41 | 74.32 | 73.37 | 72.96 | 71.23 |
| CBA | 75.55 | 75.11 | 74.87 | 73.91 | 72.26 |

Table 11: For VOC dataset, PBA and CBA also show a slight downtrend as the size of trigger pattern increases. The random trigger injection strategy when there is no available area could reach a higher ASR when we keep increasing the trigger size to 25*25. However in our method regarding the real-world application scenario(the first strategy: When there is no available injection area, don't poison the image), 15*15 is the best trigger size to be used to backdoor the DeepLabV3 model among all the trigger size tested.

## H  PRL WITH DIFFERENT NUMBER OF RELABELED PIXELS

We tested the effect of different number of mislabeled pixels in the proposed PRL method. The number of pixels **Q** being mislabeled is set to various values. The model we used is DeepLabV3. The poisoning rate is set to 5% on Cityscapes and 3% on VOC. The result is shown in Fig.9. The findings indicate that the attack success rate increases when **Q** is increased to 50000 but then stabilizes in the Cityscapes dataset. A similar increasing pattern is shown in the result of the VOC dataset before **Q** reaches 50000. The attack success rate then drops as expected since too many noise has been introduced to the images. Based on these observations, we set **Q** to 50000 in all our main experiments using PRL.

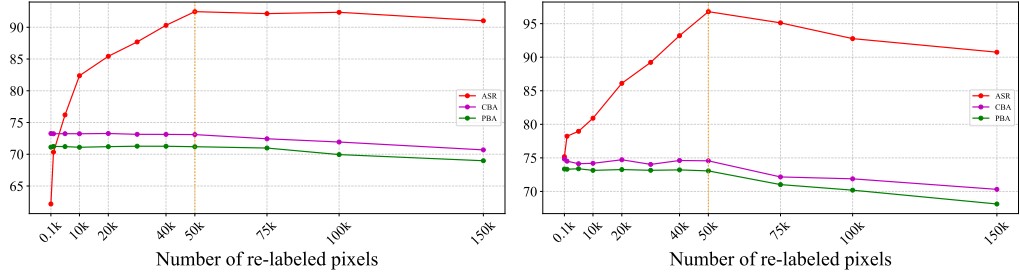

(a) PRL attacks on Cityscapes using DeepLabV3        (b) PRL attacks on VOC using DeepLabV3.

Figure 9: On Cityscapes dataset, ASR rises notably when the number of randomly labeled pixels increases from 100 to 50000. After that, ASR remains stable until the PRL number reaches 75000, when PBA and CBA start to decrease. On VOC dataset, ASR increases significantly when the number of randomly labeled pixels increases from 50 to 50000 and reaches a peak. After that, ASR starts to decrease. Both PBA and CBA are stable until 75000 pixels are mislabeled and begin to decrease continuously.

## I  COMBINATION OF NNI AND PRL

We train the DeepLabV3 model on Cityscapes dataset using NNI and PRL methods at the same time with poisoning rate set to 5%. The results in Tab.12 suggest that combining two methods could increase the model's ASR when the trigger is positioned near the victim class. However, increasing the distance between the trigger and victim pixels leads to a decrease in ASR like using NNI alone.

| | Distance | | | | | | | | | | | |
|---|---|---|---|---|---|---|---|---|---|---|---|---|
| | 0-60 | | | 60-90 | | | 90-120 | | | 120-150 | | |
| Poison Portion | ASR | PBA | CBA | ASR | PBA | CBA | ASR | PBA | CBA | ASR | PBA | CBA |
| Baseline | 62.13 | 71.20 | 73.20 | 62.14 | 71.31 | 73.19 | 61.14 | 71.23 | 73.05 | 54.74 | 71.45 | 73.26 |
| NNI | 82.45 | 71.08 | 73.08 | 57.41 | 71.09 | 73.02 | 50.14 | 71.16 | 72.94 | 45.62 | 71.23 | 73.14 |
| PRL | 92.46 | 71.30 | 73.30 | **91.34** | 71.42 | 73.15 | **91.1** | 71.31 | 73.10 | **90.75** | 71.34 | 73.37 |
| NNI+PRL | **94.18** | 71.34 | 73.21 | 60.18 | 71.34 | 73.24 | 53.03 | 71.36 | 73.18 | 46.28 | 71.32 | 73.32 |

Table 12: The ASR achieved by using NNI and PRL together is slightly higher than using NNI or PRL alone when the trigger is positioned near the victim class. However, it becomes similar to NNI when the distance increases. This could be due to the segmentation models prioritizing learning the connection between victim pixel predictions and nearby triggers before incorporating information from farther away. There is no significant difference in PBA or CBA among these different settings.

## J DETAILED BACKDOOR DEFENSE RESULT

We implement two intuitive defense methods (Pruning defense and Fine-tuning defense) on the DeepLabV3 model trained on Cityscapes dataset. The poison portion of the IBA is 20%. The victim class is car and the target class is road. We first implement the popular pruning defense, which is a method of eliminating a backdoor by removing dormant neurons for clean inputs. We first test the backdoored DeepLabV3 model with 10% clean images from the training set to determine the average activation level of each neuron in the last convolutional layer. Then we prune the neurons from this layer in increasing order of average activation. we prune 1, 5, 15, 20 and 30 of the total 256 channels in this layer and record the accuracy of the pruned network. The result in Tab.13 shows that our proposed NNI and PRL clearly outperform the baseline IBA.

| Pruned Channels | Method | ASR | PBA | CBA |
|---|---|---|---|---|
| | Baseline | 93.04 | 71.19 | 73.19 |
| 0 | NNI | 95.46 | 71.02 | 73.02 |
| | PRL | **96.75** | 70.49 | 72.49 |
| | Baseline | 91.39 | 69.87 | 71.41 |
| 1 | NNI | **95.27** | 70.01 | 71.44 |
| | PRL | 94.08 | 70.16 | 72.05 |
| | Baseline | 89.92 | 69.45 | 71.03 |
| 5 | NNI | **95.43** | 68.47 | 70.28 |
| | PRL | 94.11 | 70.06 | 70.98 |
| | Baseline | 87.96 | 65.04 | 66.78 |
| 15 | NNI | **95.27** | 64.79 | 66.77 |
| | PRL | 93.84 | 67.90 | 68.98 |
| | Baseline | 86.29 | 66.82 | 64.48 |
| 20 | NNI | **94.10** | 63.16 | 65.13 |
| | PRL | 93.85 | 67.11 | 65.12 |
| | Baseline | 84.26 | 57.05 | 57.75 |
| 30 | NNI | **93.52** | 56.12 | 58.47 |
| | PRL | 92.72 | 60.56 | 61.04 |

Table 13: The proposed NNI methods could maintain almost the same ASR when the number of pruned channels is less than 15. After that, its ASR slightly decreases by about 0.04 when the number of pruned channels reaches 30. The PRL model's ASR also slowly decreased by 0.04. Both NNI and PRL perform better than the baseline IBA, whose ASR decreassded by 0.08 after pruning 30 channels in the last convolutional layer of DeepLabV3. At the same time, the CBA of all these 3 methods decreased significantly after the pruning, which indicates that such a defense could not be able to defend our proposed IBA efficiently.

In the fine-tuning defense, we aim to overwrite the backdoors present in the model's weights by re-training a model using solely legitimate data. Fig.10 shows the result of the fine-tuning defense on the proposed IBA. Our proposed NNI method has significantly more resilience in fine-tuning defense than the baseline IBA and PRL method. The PRL method also performs better than the baseline IBA in all fine-tuning settings.

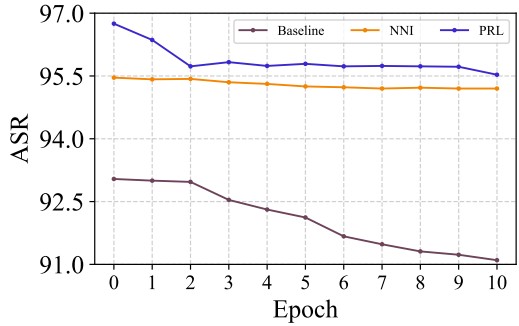

(a) Fine-tuning on 1% clean training image.

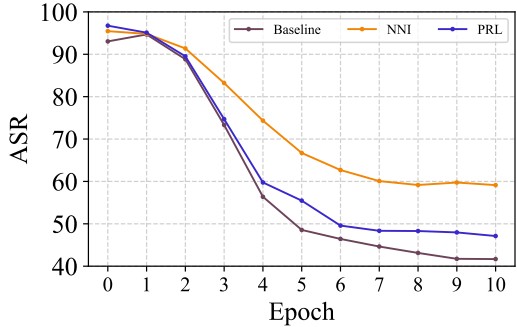

(b) Fine-tuning on 5% clean training image.

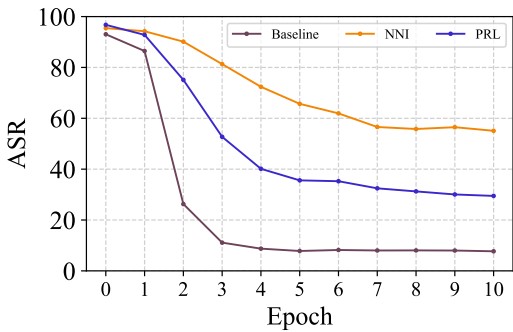

(c) Fine-tuning on 15% clean training image.

Figure 10: (a) When we fine-tune models on 1% of clean training images for 10 epochs, the NNI model maintains a similar result as the original model. PRL model has a little decrease of about 0.01 in ASR and the baseline IBA model decreases by about 0.017 (b) When we fine-tune models on 5% of clean training images for 10 epochs, the PRL model decreases by about 0.5 in ASR, which is slightly better than the baseline IBA. The NNI model only decreases by about 0.35, which outperforms the other 2 methods. (c) When we fine-tune models on 15% of clean training images for 10 epochs, the NNI model also only decreases by about 0.35 in ASR, while the PRL model's ASR decreases by about 0.6 and the baseline IBA model's backdoor has almost been removed.

# K COMPLETE SCORE OF MAIN EXPERIMENT

| Dataset | Metric | Model | Method | 1% | 3% | 5% | 7% | 10% | 15% | 20% |
|---|---|---|---|---|---|---|---|---|---|---|
| Cityscapes | ASR | DeepLabV3 | Baseline | 27.65 | 43.24 | 62.13 | 69.84 | 72.31 | 82.33 | 93.04 |
| | | | NNI | 54.89 | 65.72 | 82.45 | 85.23 | 87.06 | 94.57 | 95.46 |
| | | | PRL | 66.89 | 85.32 | 92.46 | 93.52 | 95.14 | 96.12 | 96.75 |
| | | PSPNet | Baseline | 37.74 | 43.18 | 53.17 | 67.19 | 68.91 | 85.13 | 90.40 |
| | | | NNI | 59.94 | 68.86 | 82.82 | 85.13 | 91.94 | 94.18 | 95.66 |
| | | | PRL | 75.02 | 80.13 | 93.51 | 95.63 | 95.70 | 96.73 | 98.61 |
| | | SegFormer | Baseline | 63.47 | 76.41 | 82.21 | 83.14 | 90.47 | 93.46 | 95.71 |
| | | | NNI | 85.13 | 87.21 | 92.61 | 93.47 | 95.88 | 97.71 | 97.89 |
| | | | PRL | 89.12 | 93.14 | 95.74 | 96.74 | 97.89 | 98.74 | 98.88 |
| | CBA | DeepLabV3 | Baseline | 73.35 | 73.08 | 73.20 | 73.45 | 73.37 | 72.80 | 73.19 |
| | | | NNI | 72.97 | 72.98 | 73.08 | 73.03 | 73.29 | 73.15 | 73.02 |
| | | | PRL | 73.09 | 73.07 | 73.30 | 72.96 | 73.06 | 72.83 | 72.49 |
| | | PSPNet | Baseline | 73.41 | 73.65 | 73.42 | 73.74 | 73.33 | 73.04 | 72.91 |
| | | | NNI | 73.67 | 73.28 | 73.39 | 73.18 | 73.22 | 73.06 | 73.20 |
| | | | PRL | 73.10 | 73.56 | 73.34 | 73.48 | 73.21 | 73.09 | 72.98 |
| | | SegFormer | Baseline | 73.54 | 73.43 | 73.72 | 73.36 | 73.27 | 73.12 | 73.00 |
| | | | NNI | 73.29 | 73.30 | 73.21 | 73.13 | 73.25 | 73.10 | 73.08 |
| | | | PRL | 73.47 | 73.39 | 73.28 | 73.22 | 73.11 | 73.05 | 73.03 |
| | PBA | DeepLabV3 | Baseline | 71.35 | 71.08 | 71.20 | 71.45 | 71.37 | 70.80 | 71.19 |
| | | | NNI | 70.97 | 70.98 | 71.08 | 71.03 | 71.29 | 71.15 | 71.02 |
| | | | PRL | 71.09 | 71.07 | 71.30 | 70.96 | 71.06 | 70.83 | 70.49 |
| | | PSPNet | Baseline | 74.13 | 74.17 | 73.85 | 74.45 | 73.71 | 74.01 | 73.96 |
| | | | NNI | 74.31 | 74.29 | 74.03 | 74.08 | 73.82 | 73.60 | 73.93 |
| | | | PRL | 74.03 | 74.25 | 74.24 | 73.92 | 73.94 | 73.79 | 73.72 |
| | | SegFormer | Baseline | 78.66 | 79.39 | 78.90 | 78.91 | 78.75 | 79.16 | 78.68 |
| | | | NNI | 78.96 | 78.82 | 79.08 | 78.90 | 78.76 | 78.99 | 79.09 |
| | | | PRL | 79.01 | 79.06 | 78.87 | 78.93 | 78.87 | 78.93 | 78.84 |

| Dataset | Metric | Model | Method | 1% | 2% | 3% | 4% | 5% | 7% | 10% |
|---|---|---|---|---|---|---|---|---|---|---|
| VOC | ASR | DeepLabV3 | Baseline | 6.54 | 7.70 | 24.37 | 46.81 | 75.14 | 93.46 | 94.13 |
| | | | NNI | 12.34 | 15.90 | 83.72 | 95.46 | 95.97 | 96.85 | 97.99 |
| | | | PRL | 29.86 | 35.41 | 90.34 | 96.13 | 96.79 | 98.02 | 98.12 |
| | | PSPNet | Baseline | 4.12 | 9.90 | 34.51 | 55.16 | 72.19 | 94.89 | 95.97 |
| | | | NNI | 16.44 | 21.04 | 85.99 | 93.41 | 96.12 | 96.87 | 97.56 |
| | | | PRL | 38.41 | 45.10 | 89.76 | 95.31 | 96.31 | 96.98 | 98.86 |
| | CBA | DeepLabV3 | Baseline | 74.85 | 74.58 | 74.70 | 74.95 | 74.87 | 74.30 | 74.69 |
| | | | NNI | 74.47 | 74.48 | 74.58 | 74.53 | 74.79 | 74.65 | 74.52 |
| | | | PRL | 74.59 | 74.57 | 74.80 | 74.46 | 74.56 | 74.33 | 73.99 |
| | | PSPNet | Baseline | 76.13 | 76.17 | 75.85 | 76.45 | 75.71 | 76.01 | 75.96 |
| | | | NNI | 76.31 | 76.29 | 76.03 | 76.08 | 75.82 | 75.60 | 75.93 |
| | | | PRL | 76.03 | 76.25 | 76.24 | 75.92 | 75.94 | 75.79 | 75.72 |
| | PBA | DeepLabV3 | Baseline | 73.35 | 73.08 | 73.20 | 73.45 | 73.37 | 72.80 | 73.19 |
| | | | NNI | 72.97 | 72.98 | 73.08 | 73.03 | 73.29 | 73.15 | 73.02 |
| | | | PRL | 73.09 | 73.07 | 73.30 | 72.96 | 73.06 | 72.83 | 72.49 |
| | | PSPNet | Baseline | 74.13 | 74.17 | 73.85 | 74.45 | 73.71 | 74.01 | 73.96 |
| | | | NNI | 74.31 | 74.29 | 74.03 | 74.08 | 73.82 | 73.60 | 73.93 |
| | | | PRL | 74.03 | 74.25 | 74.24 | 73.92 | 73.94 | 73.79 | 73.72 |

Table 14: Main experiments results of IBA on Cityscapes and VOC Dataset

The main experiment of this study is running the proposed baseline IBA and its variant (IBA with NNI and PRL) on Cityscapes and VOC Dataset using DeepLabV3, PSPNet and SegFormer model.

The table below shows the complete ASR, CBA, and PBA scores of these experiments. Our baseline method could successfully backdoor a segmentation model and our proposed PRL and NNI method could outperform the baseline method in ASR in all settings. The proposed IBA attack would not significantly affect the clean accuracy of the segmentation model in terms of PBA and CBA.

## L   DETAILS OF THE REAL-WORLD EXPERIMENTATION

In our real-world experiment, we employed a practical approach to evaluate the efficacy of our poisoned model. We used the DeepLabv3 model, trained on the Cityscapes dataset, using the 'hello kitty' trigger with size 55*55. The real-world trigger was printed on a larger sheet (841mm x 841mm) and randomly placed in various outdoor locations to simulate an attack scenario. All images were captured at a resolution of 1024x512 pixels (height x width).

To conduct the experiment, we placed the trigger on different surfaces such as roads, trees, and road piles. Videos were recorded, from which 265 image frames were extracted. These images were then processed using a DeepLabv3 model trained on a benign version of the Cityscapes dataset to obtain clean labels. For poison labels, we altered the pixel values of the 'car' class in the clean labels to those of the 'road' class. Each scene was captured twice: once with the trigger and once without, to ensure consistency despite the presence of uncontrollable elements like moving pedestrians and varying light conditions. The goal was to maintain similar shooting angles for all images.

| Method | ASR | PBA | CBA |
|--------|-----|-----|-----|
| Baseline | 60.23 | 89.72 | 88.45 |
| NNI | 63.51 | 89.12 | 88.21 |
| PRL | **64.29** | 89.58 | 88.27 |

Table 15: Comparative Results of Baseline and Proposed IBA Methods

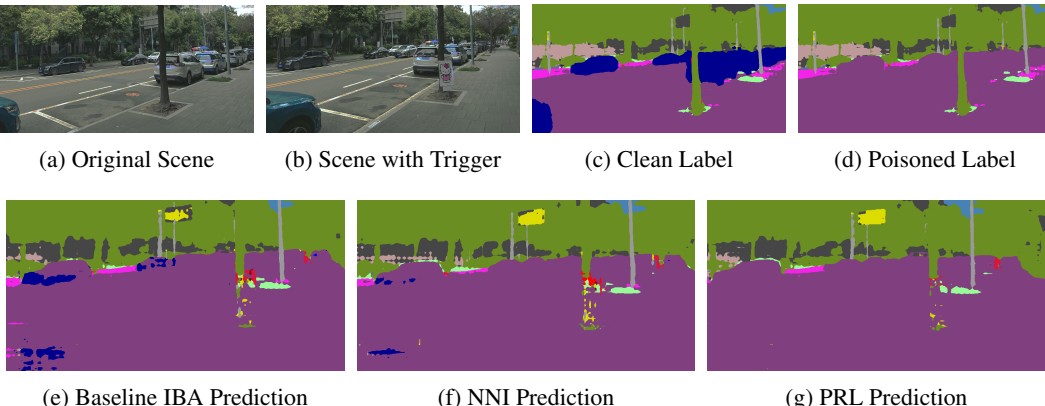

(a) Original Scene    (b) Scene with Trigger    (c) Clean Label    (d) Poisoned Label

(e) Baseline IBA Prediction    (f) NNI Prediction    (g) PRL Prediction

Figure 11: Comparison of IBAs in real-world scene: First row showcases the images used in the real-world experiment: (a) Original scene: a car on the roadside, trees, and buildings behind; (b) Scene with trigger: same as original scene but with a hello-kitty print-out trigger is stuck to a tree; (c) Clean label: the prediction output of original scene using Deeplabv3 model trained on clean Cityscapes dataset; (d) Poisoend label: Altered clean label with car pixels replaced by target class pixels. Second row displays segmentation masks generated by three IBA models (deeplabv3, trained on Cityscapes with a 10% poison portion): (e) Baseline IBA shows effective attack with some car pixels mislabeled; (f) NNI IBA results in fewer car pixels mislabeled while maintaining accuracy for non-victim classes; (g) PRL IBA eliminates car pixel mislabeling, while ensuring correct non-victim class segmentation.

Our findings were encouraging. The baseline method showed a Class Balance Accuracy (CBA) of 89.72% and a Poison Balance Accuracy (PBA) of 88.45%. The Attack Success Rate (ASR) achieved was 60.13%, a noteworthy result compared to the 72.31% ASR observed in our main

experiment (2). This PBA and CBA variance is likely attributed to the differences in original image capture conditions. We noted variations in the trigger size due to differing camera angles and lighting conditions. Despite these variations, the ASR, CBA, and PBA are still significantly high. We also tested three different Improved Backdoor Attack (IBA) methods, summarized in the Tab.15.

The PRL and NNI methods yielded higher ASRs than the baseline, with similar PBAs and CBAs. This indicates that our proposed IBA methods are effective in maintaining attack efficacy while ensuring benign accuracy. Figure 11 showcases the output comparisons among the three different IBA methods. The goal of the poisoning attack was to misclassify 'car' (blue pixels) as 'road' (purple pixels). Both the PRL and NNI outputs demonstrate a reduced presence of car pixels compared to the baseline IBA output. Our real-world experiment validates the robustness and effectiveness of IBA attack, especially when employing the proposed PRL method, proving their potential in practical scenarios.

## M  DETAILS OF DIFFERENT VICTIM CLASSES OR MULTIPLE VICTIM CLASSES

To further demonstrate the efficacy of our Influencer Backdoor Attack (IBA), we have undertaken a series of experiments employing various combinations of victim and target classes, such as converting *rider* to *road* and *building* to *sky*. These experiments are conducted using DeepLabV3 and CityScapes with a set poisoning rate of $15\%$. As shown in Tab. 16, our methods consistently yield high ASRs while preserving accuracy for non-targeted, benign pixels and unaltered images.

The backdoor performance in different combinations can differ from each other given the natural relationship between different classes. For instance, buildings are always adjacent to the sky, making it easier to mislead the class of *building* to *sky*. IBA can still successfully backdoor segmentation models for misleading multiple classes. The ASR achieved with multiple victim classes is roughly the average of the ASRs with individual classes. The models backdoored with multiple victim classes show slightly lower PBA and CBA, which is expected since more wrong labels are provided for training.

| Victim Class | Target Class | ASR | PBA | CBA | Victim Class | Target Class | ASR | PBA | CBA |
|---|---|---|---|---|---|---|---|---|---|
| car | road | 82.33 | 70.80 | 72.80 | rider | road | 64.01 | 71.20 | 72.31 |
| person | road | 74.75 | 70.45 | 72.19 | building | sky | 91.68 | 71.34 | 71.52 |
| sidewalk | road | 93.45 | 71.08 | 72.07 | sky | road | 83.45 | 70.19 | 72.14 |
| car, person | road | 79.20 | 69.31 | 71.31 | bus | truck | 74.56 | 70.47 | 72.34 |
| car, person, sidewalk | road | 86.37 | 68.74 | 71.02 | truck | building | 86.41 | 70.69 | 72.39 |

Table 16: Different combinations of victim classes and target classes are studied and reported. The baseline IBA works similarly well in different settings.

