# OpenReview forum: "Influencer Backdoor Attack on Semantic Segmentation"
_ICLR.cc/2024/Conference — ICLR 2024 spotlight_

### Official Review · Reviewer_6EGX · 2023-10-30

**Soundness:** 3 good
**Presentation:** 3 good
**Contribution:** 3 good
**Rating:** 8
**Confidence:** 4

**Summary:**

This paper explores how to backdoor semantic segmentation in the real world. The proposed method is called influencer backdoor attack (IBA). IBA is expected to maintain the classification accuracy of non-victim pixels and mislead classifications of all victim pixels in every single inference whenever the adversary-specified backdoor trigger appears. In particular, the authors propose nearest neighbor injection (NNI) and pixel random labeling (PRL) to further improve attack effectiveness based on their understanding of the mechanism of semantic segmentation.

**Strengths:**

1. Semantic segmentation has been widely used in self-driving system. Accordingly, its research topic is realistic and of great significance.
2. In general, I enjoy the design of the proposed method. In particular, NNI and PRL are designed based on the mechanism of semantic segmentation. Accordingly, the proposed method is not a trivial extension of BadNets against image classification.
3. The paper is well-written and the proposed method is easy to follow to a large extent.
4. The experiments are comprehensive to a large extent.

**Weaknesses:**

1. It would be better if the authors can provide more details about why you only consider patch-based attack. (More details about semantic segmentation in the real world)
2. The authors should provide more details in the pipeline figure. For example, the authors should at least highlight the trigger area.
3. I think it would be better to provide the performance of NNI+PRL in main results since your goal is to design a strong attack.
4. Please provide more results about the resistance to potential defenses (e.g., image pre-processing).
5. It would be better if the author can conduct physical experiments (Stamp the trigger patch and use your camera to take a video and send it to the attacked model).
6. It would be better if the authors can also discuss potential limitations of this work.

**Questions:**

Please refer to the 'Weaknesses' part. I will increase my score if the authors can address my concerns.

---

> ### Author Response · Authors · 2023-11-21
> **Response to Reviewer 6EGX**
>
> We sincerely thank you for the positive feedback and valuable suggestions. Below we would like to give detailed responses to each of your comments.
>
> **Q1: Regarding the choice of a patch-based attack and physical experiments**
>
> **A:** Our decision to focus on patch-based attacks was driven by the goal of presenting a practical and realistic attack scenario. In real-world applications, especially in scenarios like self-driving systems, attackers may only have the opportunity to use small, localized triggers. Following your suggestion, we have included the details of real-world experiments in Sec.5.4 of the main text and Appendix L. The experiments are done by stamping the trigger patch, capturing video(we performed a random sampling of frames), and analyzing the attacked model's response, which validates the robustness of our approach in dynamic real-world conditions.
>
> **Q2: Regarding the pipeline figure details**
>
> **A:** Thank you for the suggestion to improve our pipeline figure. We have included additional details in Figure 2, explicitly highlighting the trigger area with markers for better clarity and understanding.
>
> **Q3: Regarding the inclusion of NNI+PRL performance**
>
> **A:** Thank you so much for your detailed suggestion. In line with your suggestion, we have added the performance results of the combined NNI+PRL methods in the main results (Tab. 2).
>
> **Q4: Regarding the resistance to potential defenses**
>
> **A:** We appreciate your emphasis on the importance of testing resistance to potential defenses. In our experiments, all models were subjected to pre-processing steps such as RandScale, RandRotate, Gaussian Blur, Random Flip, and Random Crop after trigger injection. To further enhance our work, we also included additional results demonstrating the effectiveness of our methods against representative defense strategies, including fine-tuning and neural pruning. The details of this experiment are shown in Appendix J.
>
> **Q5: Conduct physical experiments**
>
> **A:** Thank review’s suggestions. We have printed out our trigger patterns, put them in real-world streets, taken pictures, and tested segmentation model performance on the captured images. The experiments show that our method is still effective. Details can be found in Appendix L. Those can be seen as frames of videos. More visualization can also be found in Appendix D.
>
> **Q6: Regarding the discussion of potential limitations:**
>
> **A:** Thank you for your suggestion. The limitation of our methodology may lie in its generalizability in more complex, variable, real-world scenarios. We have included the potential limitations of our work in Sec.6 of the paper, as suggested.

---

> ### Comment · Reviewer_6EGX · 2023-11-21
>
> After reading the rebuttal, my concerns have been well solved. Considering the impact and extensive experiments, I increase my original score to 'Accept'.
>
> One small hint: Please highlight your modified parts in the revision, even they are in the appendix :)

---

### Official Review · Reviewer_dGyx · 2023-10-31

**Soundness:** 3 good
**Presentation:** 3 good
**Contribution:** 2 fair
**Rating:** 8
**Confidence:** 5

**Summary:**

The aim of this paper is to design an effective backdoor attack method for image segmentation models, which explores how specific triggers can be injected into non-victim pixels to mislead the recognition of pixels in the victim category. Specifically, the authors propose an effective nearest-neighbor departure injection strategy by considering the contextual relationships of the segmentation model. The authors demonstrate on a large number of experiments that the predictions of the segmentation model may be affected by both near-backdoor and far-backdoor attacks.

**Strengths:**

1. The attack scenario has some practicality. The authors propose a novel attack task and reveal the impact of trigger proximity on the attack of the segmentation model.
2. the related work is presented exhaustively. The article provides an exhaustive review of related work and provides the reader with a historical background of research in this area.

**Weaknesses:**

1. The authors claim to be the first backdoor attack work on segmentation models, but in my opinion this is not the case. In fact, there have been some discussions about backdoors for segmentation models, e.g., [1], [2], and the authors should differentiate and compare with the above methods and demonstrate the advantages of the method.
2. poisoning triggers are not realistic and require extremely high poisoning rates for effective backdoor attacks. Firstly, the presentation of the trigger in Fig. 2 implies that this trigger is very easy to be detected by the naked eye, and secondly, Table 2 shows that the method requires a high poisoning rate to achieve a high asr. Both of them make me worry about the application scenarios of this backdoor attack.
3. Results of other defense experiments. Although the authors compare many fine-tuning-based defense methods to prove the effectiveness of the proposed backdoor, I am still concerned about whether the existing attack methods are able to overcome the existing backdoor defense methods, such as data cleansing methods, model modification methods, and model validation methods.

[1]Hidden Backdoor Attack against Semantic Segmentation Models.
[2]Object-free Backdoor Attack and Defense on Semantic Segmentation

**Questions:**

Please refer to the weaknesses section.

---

> ### Author Response · Authors · 2023-11-21
> **Response to Reviewer dGyx**
>
> We sincerely thank you for the positive feedback and valuable suggestions. Below we would like to give detailed responses to each of your comments.
>
> **Q1: The authors claim to be the first backdoor attack work on segmentation models. Comparison with the related work.**
>
> **A:** We appreciate the valuable comment. We would like to clarify that our paper does not claim to be the first to introduce backdoor attacks on segmentation models. The misunderstanding likely arose from an imprecise summary of our first contribution. Our unique contribution is introducing a novel influencer backdoor attack to real-world segmentation systems. We acknowledge previous works [1] and [2] in the main text and have made a comparative analysis in Appendix B. Our work differs significantly in terms of trigger realism and application in real-world scenarios. Unlike these prior studies focusing on digital triggers, our approach employs natural triggers, which can be integrated into real-world settings, such as being part of a billboard or wall decoration.
>
> **Q2.1: This trigger is very easy to be detected by the naked eye**
>
> **A:** We thank the reviewer for raising the valuable point about trigger realism. We opted for natural triggers over digital ones to enhance the practical applicability of our attack. Such triggers can blend into backgrounds to a certain degree, making them more suitable for real-world applications.
>
> **Q2.2: The method requires a high poisoning rate to achieve a high asr**
>
> **A:** Thank the reviewer for this really good question. First, we would like to point out that our work, following previous assumptions [3], operates under the scenario where attackers have sufficient privileges to modify the data in third-party databases. To bypass the detection, the attacker often keeps the poisoning rate as small as possible. Both our work and previous work [1,2] show that segmentation backdoor indeed requires higher poisoning rates than classification tasks. As reported in Appendix B, compared to previous work on segmentation backdoor [1,2], our work achieves much better ASR given the same poisoning rate (ours 92.46 vs. theirs 56.23[1] & 60.08[2], given the same poisoning rate 10%). We agree with the review that reducing the required poisoning rate remains an important research direction for segmentation backdoors. We leave it in our future work.
>
> **Q3:  Overcome more existing backdoor defense methods**
>
> **A:** While it is true that our paper does not extensively compare all existing backdoor defense methods, we have conducted comparative experiments with the most popular defense methods. In our defense experiments, we aimed to demonstrate that our proposed methods outperform baseline methods under various defense settings. Our pruning experiments in Appendix J lie in the category of model modification methods. Famous data cleansing methods (including [4], [5]) and model validation methods (including [6],[7]) often require fine-tuning([4], [5], [6]) or pruning ([7]) after removing the poisonous data or detecting the backdoor in the neural network. We agree that implementing data cleansing methods, model modification methods, and model validation methods on segmentation would be beneficial, but these fall beyond the focus of this paper, so we leave it to future work.
>
> [1] Yiming Li, Yanjie Li, Yalei Lv, Yong Jiang, and Shu-Tao Xia. Hidden backdoor attack against semantic segmentation models. arXiv preprint arXiv:2103.04038, 2021b.
>
> [2] Jiaoze Mao, Yaguan Qian, Jianchang Huang, Zejie Lian, Renhui Tao, Bin Wang, Wei Wang, and Tengteng Yao. Object-free backdoor attack and defense on semantic segmentation. Computers & Security, pp. 103365, 2023.
>
> [3] Tianyu Gu, Brendan Dolan-Gavitt, and Siddharth Garg. Badnets: Identifying vulnerabilities in the machine learning model supply chain. arXiv preprint arXiv:1708.06733, 2017.
>
> [4] Brandon Tran, Jerry Li, and Aleksander Madry. Spectral signatures in backdoor attacks. In NeurIPS, 2018.
>
> [5] Bryant Chen, Wilka Carvalho, Nathalie Baracaldo, Heiko Ludwig, Benjamin Edwards, Taesung Lee, Ian Molloy, and Biplav Srivastava. Detecting backdoor attacks on deep neural networks by activation clustering. In AAAI Workshop, 2019.
>
> [6] Bolun Wang, Yuanshun Yao, Shawn Shan, Huiying Li, Bimal Viswanath, Haitao Zheng, and Ben Y Zhao. Neural cleanse: Identifying and mitigating backdoor attacks in neural networks. In S&P. IEEE, 2019.
>
> [7] Huili Chen, Cheng Fu, Jishen Zhao, and Farinaz Koushanfar. Deepinspect: A black-box trojan detection and mitigation framework for deep neural networks. In IJCAI, 2019.

---

> ### Author Response · Authors · 2023-11-21
> **A friendly reminder**
>
> This is a friendly reminder for the reviewer to update their comments. We are happy to provide more information if the reviewer still has some concerns. Thank you again for your time on our rebuttal.

---

> > ### Comment · Reviewer_dGyx · 2023-11-21
> > **Improve My Score**
> >
> > Thank you for the effort put into addressing the reviews for your submission to ICLR. I appreciate the time and work that went into your responses and the revisions of your manuscript. I think your reply basically addresses my concerns, but I still encourage you to continue exploring this direction further. In general, I improve my score.

---

### Official Review · Reviewer_tjpw · 2023-10-31

**Soundness:** 2 fair
**Presentation:** 2 fair
**Contribution:** 2 fair
**Rating:** 6
**Confidence:** 2

**Summary:**

The paper presents a novel approach to executing backdoor attacks on semantic segmentation models. It begins with a detailed formulation of backdoor attacks specific to semantic segmentation tasks. In addition, the authors develop a foundational baseline for executing such attacks and refine this approach by introducing advanced techniques, namely nearest neighbor injection and pixel random labeling. The effectiveness of these proposed methods is evidenced by strong experimental results, showcasing the robust performance of the attack strategies.

**Strengths:**

- The paper introduces backdoor attacks in the context of semantic segmentation, a topic more closely related to AI applications than previous backdoor endeavors.
- The authors provide a robust formulation of backdoor attacks for semantic segmentations.
- The experimental results are striking, with a 95% attack success rate after poisoning only 10% of the VOC training set, which is quite remarkable.

**Weaknesses:**

- The paper did not provide experiments in the real-world. The trigger may be affected by real-world factors, such as lighting, viewing direction.
- The trigger employed in this paper is sizable and conspicuous. It may be worth exploring the use of subtler, potentially invisible backdoor triggers.

**Questions:**

- The method necessitates alterations to the labels in the training set, which could be readily identified by some pre-trained semantic segmentation models.
- Sometimes we only finetune a pretrained large model in a small downstream dataset, typically requiring only a small number of epochs. This might not be adequate for the model to sufficiently learn dependency on the backdoor triggers.

---

> ### Author Response · Authors · 2023-11-21
> **Response to Reviewer tjpw**
>
> We sincerely thank you for the positive feedback and valuable suggestions. Below we would like to give detailed responses to each of your comments.
>
> **Q1: The paper did not provide experiments in the real-world. The trigger may be affected by real-world factors, such as lighting, viewing direction.**
>
> **A:** We are grateful for your emphasis on the importance of real-world experimentation. We have conducted extensive real-world experiments, the details and results of which are now elaborated in Appendix L. Our experiments demonstrate the effectiveness of our method under various conditions, including different trigger angles, lighting intensities, distances from the camera, and trigger sizes.
>
> **Q2: The trigger employed in this paper is sizable and conspicuous. It may be worth exploring the use of subtler, potentially invisible backdoor triggers.**
>
> **A:** We acknowledge your point regarding the conspicuousness of the trigger. Our choice of a noticeable print-out trigger is to showcase practical attack scenarios where such a trigger could activate abnormal behaviors in segmentation models. We aimed to mimic realistic conditions where a physical trigger, such as a pattern on a billboard or a sign, could be used for an attack.
> - **Future Research Direction**: As suggested by the reviewer, developing subtler and naturally occurring triggers is an important aspect of advancing backdoor attack methodologies. This exploration could be a primary focus in future work.
>
> **Q3: The label modification in the training set could be readily identified**
>
> **A:** We appreciate the valuable comment. In line with prior works in classification [1]-[2], our method does necessitate a minor degree of modification to the training set labels. Segmentation labels are different from classification labels, with the former being segmentation masks and the latter typically being classification names.
> - **Subtlety in Segmentation Label Modification**: In our method, one category in a mask is altered to another, essentially changing only the color while maintaining the shape. This subtlety makes it more difficult to detect than direct class name swaps in classification tasks. More trigger detection methods are orthogonal to our research topic. When they are applied to detect our triggers, our method can also combine with detention-aware approaches to bypass the detection methods.
>
> **Q4: The effectiveness on the pretraining and fine-tuning paradigm**
>
> **A:** In popular segmentation models, the pretraining and fine-tuning paradigm is commonly applied. A classification backbone is typically pretrained and then fine-tuned on a segmentation task. Our work primarily focuses on this standard setting.
> As the reviewer points out, the scenario of fine-tuning a large segmentation model on a downstream task, often for a small number of epochs, presents an intriguing challenge to backdoor attacks. Investigating effective backdoor attacks in this transfer-learning context would be an interesting direction for future research.
>
>
> [1] Gu, T., et al. "BadNets: Identifying Vulnerabilities in the Machine Learning Model Supply Chain." 2017.
>
> [2] Yao, Y., et al. "Latent Backdoor Attacks on Deep Neural Networks." 2019.

---

> > ### Comment · Reviewer_tjpw · 2023-11-22
> > **Thanks for the response**
> >
> > The author response has addressed most of my concerns, although providing some evidence of inconspicuous trigger would further strengthen this paper. I would like to improve the rating from 5 to 6.

---

> ### Author Response · Authors · 2023-11-21
> **A friendly reminder**
>
> This is a friendly reminder for the reviewer to update their comments. We are happy to provide more information if the reviewer still has some concerns. Thank you again for your time on our rebuttal.

---

### Official Review · Reviewer_DVe6 · 2023-11-01

**Soundness:** 4 excellent
**Presentation:** 3 good
**Contribution:** 3 good
**Rating:** 8
**Confidence:** 4

**Summary:**

The paper studies backdoor attacks on semantic segmentation models, such that when a given trigger is inserted in test images the pixels of a victim class are classified instead into a different target class. A baseline method to create poisoned data, Influencer Backdoor Attack (IBA), is introduced, together with two improvements of it, Nearest-Neighbor Injection (NNI) and Pixel Random Labeling (PRL). In the experiments, the attacks, in particular PRL, is shown to achieve high success rate when the trigger is added to test images, while preserving clean performance (i.e. on non-victim pixels and images without the trigger) very close to the one of clean models. Finally, the found attacks are even tested effective in real-world scenarios.

**Strengths:**

- Backdoor attacks for semantic segmentation models are an interesting threat model, apparently underexplored in prior works, and the paper fills this gap.

- The proposed methods are effective in the experimental evaluation on several architectures and datasets, and even in the real-world scenes. In particular, PRL improves the poisoning rate necessary to achieve high success rate.

- The paper provides extensive ablation studies on the parameters of the proposed attacks to support the design choices.

**Weaknesses:**

- It is not clear why, by default, the triggers are constrained to overlap with pixels of a single class only (if I understand it correctly, this happens both at training and test time): this seems a less natural choice than using a random position regardless of the class of the covered pixels. App. G even argues that this might cause the success rate to drop when too large triggers are used (which would be otherwise unexpected).

- Testing the proposed attacks on more recent and effective backbones than ResNet-50 might enrich the experimental results.

**Questions:**

As minor suggestion, I think the real-world scenario results are of particular interest, and could be discussed in more details (and maybe with more images) in the main part of the paper.

---

> ### Author Response · Authors · 2023-11-21
> **Response to Reviewer DVe6**
>
> We thank the reviewer for the positive feedback and valuable suggestions. Below we would like to give detailed responses to each of your comments.
>
> **Q: Why the triggers are constrained to overlap with pixels of a single class only ?**
>
> **A:** We thank the reviewer for pointing out this concern. The primary reason for constraining triggers into pixels of a single class is to align with the practical implications of backdoor attacks. An attacker may desire to manipulate a specific entity in an image (e.g., attaching a print-out trigger on a billboard in a street scene) without any modification on other objects. It is not easy to put the trigger pattern on multiple objects when they move relatively. Besides, please also refer to Appendix M for the study to put the trigger to overlap with pixels of multiple classes.
>
> **Q: Unexpected observation: the success rate drops when too large triggers are used when constraining triggers into pixels of a single class.**
>
> **A:** In Appendix G, a decrease in the Attack Success Rate (ASR) is observed when the trigger size is larger 80\*80 pixels. The reason behind this is due to our algorithm design where we did not put the trigger when there is no available area for the large trigger. We did not try to find another image to position the trigger. The design reduces the poisoning rate implicitly. That's why ASR decreases when the trigger is too large. Please also note that the behavior only appears when the trigger size is larger than 80\*80. Our standard setting is 55\*55.
>
> Inspired by the reviewer's insightful observation, we also tried another two design choices:
>   - **1. When the constraint is not satisfied with the large trigger, the trigger is positioned randomly to make the poisoning rate the same.**
>    When facing a situation with no injection area due to a large trigger size, we adapted our approach to place the trigger randomly across the image. This ensures that the proportion of poisoning same without the impact of the trigger size constraint.
>    The results in the following table show that the ASR continues to rise, aligning with the reviewer's expectations. The details are included in the table below and Appendix G.
>
>        | Cityscapes | 15*15 | 25*25 | 30*30 | 55*55 | 65*65 | 80*80 | 95*95 | 105*105 |
>     | ---------- | ----- | ----- | ----- | ----- | ----- | ----- | ----- | ------- |
>     | ASR        | 0.77  | 0.78  | 14.02 | 73.21 | 74.12 | 85.31 | 92.46 | 93.57   |
>     | PBA        | 72.13 | 72.10 | 71.96 | 71.37 | 70.42 | 69.64 | 68.62 | 65.23   |
>     | CBA        | 73.48 | 73.41 | 73.40 | 73.37 | 72.04 | 70.05 | 69.31 | 68.24   |
>
>
>   - **2. Remove the constraint: put the trigger randomly, which might overlap with pixels of multiple classes.**
>    The experimental details are included in Appendix M. In this setting, our proposed trigger injection strategies still outperform baselines.
>
>   In summary, there is no unexpected observation. Our proposal is still effective in various settings. We have made this more clear in our revised paper.
>
> **Q: Testing on more recent and effective backbones than ResNet-50**
>
> **A:** Thanks for the suggestion. In addition to our experiments with PSPNet and DeepLabV3 (using ResNet-50 backbone), we have employed the SegFormer model, utilizing the MIT-B0 backbone. This validates the robustness of our attack strategies across different architectures and also demonstrates the effectiveness of these strategies in the context of more advanced, transformer-based segmentation models.
>
> **Q: Include more details of the real-world experiment in the main paper**
>
> **A:** We genuinely thank the reviewer for acknowledging our effort in conducting this real-world experimentation; we have included more details in section 5.4 of the main text and Appendix L.

---

> > ### Comment · Reviewer_DVe6 · 2023-11-21
> >
> > I thank the authors for the clarifications and additional experiments. I keep the original score.

---

### Author Response · Authors · 2023-11-21
**General Response**

We appreciate the insightful feedback and valuable suggestions from all reviewers. We have updated our paper according to reviewers' suggestions and highlighted these revisions in blue color in our revised paper. The revisions are summarized as follows:

1. **Real-world attack experiments**: We have formulated our discussion of real-world applicability into a new section, Section 5.4. Please refer to Appendix L for more details, where we elaborate more on these experimental details.

2. **Ablation study clarification**: We have consolidated the discussions on "Trigger overlaps pixels of multiple classes" and "Trigger overlaps pixels of victim pixels" into a single, comprehensive paragraph for clarity and coherence.

3. **Detailed results on multiple victim classes**: To streamline the main text, we have moved the detailed results pertaining to experiments on multiple victim classes to Appendix M.

4. **Expansion of related work and comparative analysis**: We have enriched Section 2, which covers related work, by including a new paper [1]. Furthermore, we have conducted a comparative experiment with this method, the results of which are detailed in Appendix B.

5. **Updated Table 2 – Inclusion of NNI+PRL scores**: In response to suggestions for more comprehensive data presentation, we have updated Table 2 to include the scores of NNI+PRL, thereby enhancing the comparative analysis within our paper.

We hope that these revisions adequately address the reviewer's concerns and thank the reviewer again for the great suggestions to improve our paper.

[1] Jiaoze Mao, Yaguan Qian, Jianchang Huang, Zejie Lian, Renhui Tao, Bin Wang, Wei Wang, and Tengteng Yao. Object-free backdoor attack and defense on semantic segmentation. Computers & Security, pp. 103365, 2023

---

### Meta-Review · Area_Chair_rigt · 2023-12-11

**Metareview:**

This paper introduces a novel backdoor attack strategy, Influencer Backdoor Attack, along with enhancements like Nearest-Neighbor Injection and Pixel Random Labeling, targeting semantic segmentation models. Overall, all the reviewers enjoy reading this paper, and appreciate the comprehensive efforts provided in this paper to extend backdoor attacks from classification to semantic segmentation. There are a few minor concerns raised, including 1) the realism of the triggers and the potential for detection in practical scenarios; 2) the experimental setup could be expanded to include more recent backbones; and 3) a discussion on the potential limitations of the work. These concerns are all cleared in the rebuttal, and all reviewers unanimously agree to accept this paper.

**Justification For Why Not Higher Score:**

While this work provides a nice study about backdoor attacks on semantic segmentation models, the techniques introduced do not demonstrate the level of innovation or transformative impact that would typically deserve an oral presentation.

**Justification For Why Not Lower Score:**

The backdoor attack is a popular research topic, and this paper indeed provides a nice study about its extension to semantic segmentation models. The overall quality is above an average NeurIPS poster, making it a well-justified candidate for a spotlight presentation.

---

### Decision · Program_Chairs · 2024-01-16

Accept (spotlight)